# The genetic architecture of human cerebellar morphology supports a key role for the cerebellum in human evolution and psychopathology
Torgeir Moberget [1,2] ✉, Dennis van der Meer[1,3,10], Shahram Bahrami [1,10], Daniel Roelfs[1],
Oleksandr Frei [1], Tobias Kaufmann [1,4], Sara Fernandez-Cabello[1], Milin Kim [1], Thomas Wolfers [4,5,6],
Joern Diedrichsen [7], Olav B. Smeland[1], Alexey Shadrin [1], Anders Dale[8], Ole A. Andreassen[1] &
Lars T. Westlye [1,9]

The functional domain of the cerebellum has expanded beyond motor control to also include cognitive and affective functions. In line with this notion, cerebellar volume has increased over recent primate evolution, and cerebellar alterations have been linked to heritable mental disorders. To map the genetic architecture of human cerebellar morphology, we here studied a large imaging genetics sample from the UK Biobank (n discovery = 27,302; n replication: 11,264) with state-of-the art neuroimaging and biostatistics tools. Multivariate GWAS on regional cerebellar MRI features yielded 351 significant genetic loci (226 novel, 94% replicated). Lead SNPs showed positive enrichment for relatively recent genetic mutations over the last 20-40k years (i.e., overlapping the Upper Paleolithic, a period characterized by rapid cultural evolution), while gene level analyses revealed enrichment for human-specific evolution over the last ~6-8 million years. Finally, we observed genetic overlap with major mental disorders, supporting cerebellar involvement in psychopathology.

The cerebellum contains ~80% of all neurons in the human brain[1] and has rapidly expanded in volume over recent primate evolution[2]. Indeed, the surface area of the cerebellar cortex extends to almost 80% of the surface area of the cerebral cortex[3]. Comparative genetic analyses suggest that protein coding genes with known roles in cerebellar development have been subject to a similar, or even greater, rate of hominid evolution as compared to cerebro-cortical developmental genes[4]. Thus, the evolution of the cerebellum may have played a key role in the emergence of human cognition, including language[5].

A growing number of neuroimaging and clinical studies in humans also link cerebellar structure and function to a wide range of cognitive and affective functions[6–8], as well as to a number of heritable developmental[9] and psychiatric[10] disorders where these abilities either fail to develop properly or are compromised later in life. However, compared to supra-tentorial brain structures such as the cerebral cortex[11] and the hippocampus[12], few studies have mapped the genetic architecture of the cerebellum.

Of note, the few existing cerebellar genome-wide association studies (GWAS) have mostly been restricted to total cerebellar volume[13,14], thus largely ignoring regional variation in cerebellar morphology. Importantly, such variation in the relative volumes of cerebellar subregions (i.e., variation in cerebellar shape independent of total cerebellar volume) has been associated with variation in behavioral repertoires in several species[15,16], including domain-general cognition in primates[16]. Further, several recent studies investigating other brain phenotypes (e.g. the cerebral cortex[17,18] and

[1]Center for Precision Psychiatry, Division of Mental Health and Addiction, Oslo University Hospital & Institute of Clinical Medicine, University of Oslo, Oslo, Norway. [2]Department of Psychology, Pedagogy and Law, School of Health Sciences, Kristiania University College, Oslo, Norway. [3]School of Mental Health and Neuroscience, Faculty of Health, Medicine and Life Sciences, Maastricht University, Maastricht, The Netherlands. [4]Department of Psychiatry and Psychotherapy, Tübingen Center for Mental Health, University of Tübingen, Tübingen, Germany. [5]Department of Psychology, Friedrich Schiller University of Jena, Jena, Germany. [6]German Center for Mental Health (DZPG), partner site Halle/Jena/Magdeburg, Magdeburg, Germany. [7]Brain and Mind Institute, University of Western Ontario, London, Canada. [8]Center for Multimodal Imaging and Genetics, University of California at San Diego, La Jolla, CA, USA. [9]Department of Psychology, Faculty of Social Sciences, University of Oslo, Oslo, Norway. [10]These authors contributed equally: Dennis van der Meer, Shahram Bahrami.
✉e-mail: torgeir.moberget@gmail.com

the hippocampus[19]) have shown that genetic signals tends to be distributed across brain regions, rendering multivariate methods more sensitive than univariate methods.

To map the genetic architecture of human cerebellar morphology, we here analyzed a large population-based sample from the UK Biobank (*n* discovery = 27,302; *n* replication: 11,264) using state-of-the art neuroimaging and biostatistics tools. Multivariate GWAS on regional cerebellar MRI features yielded 351 significant genetic loci (228 novel, 94% replicated). Lead SNPs showed positive enrichment for relatively recent genetic mutations over the last 20–40 k years (i.e., overlapping the Upper Paleolithic, a period characterized by rapid cultural evolution), while gene level analyses revealed enrichment for human-specific evolution over the last ~6–8 million years. Finally, we observed genetic overlap with major mental disorders, supporting cerebellar involvement in psychopathology.

## Results

### Data-driven decomposition of cerebellar grey matter maps reveals highly reproducible structural covariance patterns

Since traditional atlases of the cerebellar cortex based on gross anatomical landmarks (i.e., lobules) only partially overlap with more recent functional parcellations of the cerebellum[20–28], we first used a data-driven approach (non-negative matrix factorization, NNMF[29–31]) to parcellate voxel-based morphometry (VBM) based maps of cerebellar grey matter volume (containing 147,121 $1 \times 1 \times 1$ mm voxels) from 28,212 participants into a smaller number of structural covariance patterns (SCPs), i.e., cerebellar subregions where voxel-volumes co-vary consistently across individuals (see Online Methods for details about the study sample, MRI processing and quality control procedures). Like other dimensionality reduction methods, such as principal component analysis (PCA) or independent component analysis (ICA), NNMF decomposes the input matrix (here 147k voxels × 28k participants) into two lower rank matrices (voxel weights: W, and participant weights: H), so that the product of W and H approximates the original input data. The defining feature of NNMF is that it requires these lower rank matrices to be non-negative, which has previously been shown to result in sparse, reproducible and easily interpretable parcellations of high-dimensional brain imaging data[29–33]. In essence, when applied to voxel-wise indices of grey matter volume, NNMF yields distinct maps of voxels that show similar patterns of volume-variation across individuals, commonly referred to as "structural covariance patterns"[29,30,32]. For each structural covariance pattern (common across all participants), NNMF also provides

individual subject weights expressing the degree to which these patterns are expressed, i.e., reflecting individual variation in regional cerebellar volumes.

In the current study, NNMF yielded highly reproducible cerebellar structural covariance patterns (across split-half datasets) for model orders (i.e., number of components/patterns specified) ranging from 2 to 100 (see Supplementary Fig. 2 for summary maps of all tested model orders). After observing that the improved fit to the original data seen with higher model orders tended to level off between 15 and 30 components (indicating that the intrinsic dimensionality of the data might have been reached (see Supplementary Fig. 3), we decided on a model order of 23 based on its good split-half reproducibility (see Fig. 1b, and Online Methods for details regarding model order selection). Importantly, when subject weights were summed across the 23 structural covariance maps, the resulting values corelated tightly ($r = 0.9995$) with estimates of total cerebellar grey matter volume (see also Supplementary Fig. 4), demonstrating that our data-driven decomposition preserved inter-individual variation in cerebellar volume.

Of note, our data-driven decomposition differed markedly from the standard cerebellar atlas based on gross anatomical features, shown as dotted lines in Fig. 1a (see Supplementary Fig. 3 for all 23 components and Supplementary Data 1 for quantification of overlap between NNMF-derived structural covariance patterns (SCPs) and standard cerebellar anatomical regions, i.e., lobules). For instance, five distinct SCPs overlapped Crus I of cerebellar lobule VII (shown in Fig. 1b), an anatomical region that already started to split into separate components at a model order of three. We further observed only partial overlap with task-based functional parcellations of the cerebellar cortex (Supplementary Data 2). While some SCPs clearly overlapped cerebellar regions previously associated with hand movements, eye-movements/saccades or autobiographical recall, other data-driven SCPs overlapped multiple functionally defined cerebellar regions Supplementary Data 2).

### Cerebellar structural covariance patterns are heritable and reveals a distinct anterior-posterior pattern based on their bivariate genetic correlations

After removal of one of each genetically related pair of individuals ($n = 910$), 27,302 participants remained for the genetic analyses. In addition to the 23 regional cerebellar structural covariance patterns of primary interest, we also included total cerebellar volume, estimated total intracranial volume and 9 cerebral brain phenotypes to serve as covariates and/or comparison phenotypes. Prior to all genetic analyses, morphological features were adjusted

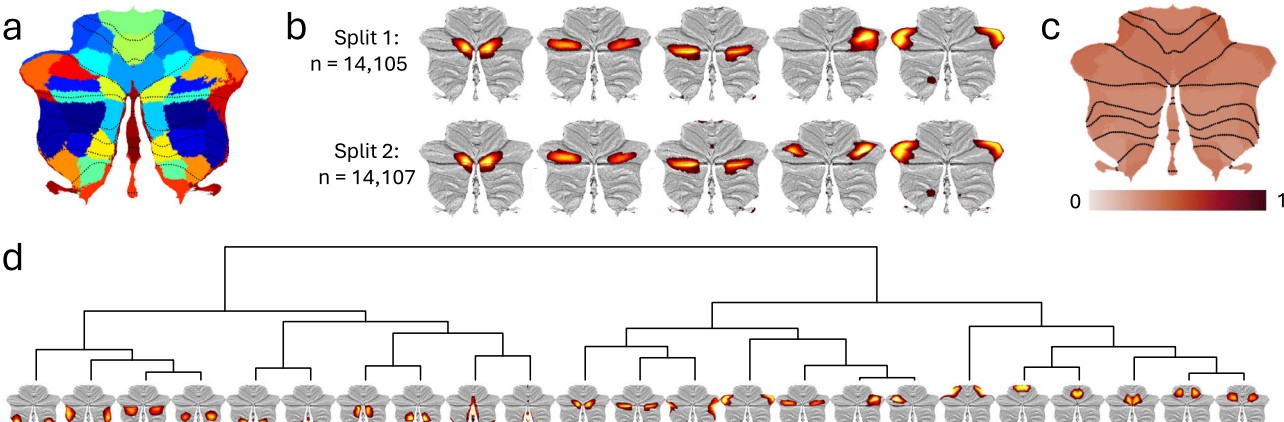

**Fig. 1 | Data driven decomposition of cerebellar grey matter maps yields highly reproducible and moderately heritable structural covariance patterns (SCPs). a** Binarized winner-takes-all map for the 23-component solution based on data-driven decomposition of cerebellar grey matter maps from 28,212 participants. Note that empirically derived boundaries between cerebellar regions only partially follow traditional lobular borders (marked with dotted black lines); **b** Five distinct components overlapping cerebellar Crus I derived from the split-half reliability analyses. While one of these SCPs emerged as bilateral in Split 2, the remaining four SCPs were

almost identical, despite being derived from two independent samples; **c** Narrow sense (SNP-based) heritability of the 23 SCPs (see Supplementary Data 5 for numerical values). **d** Hierarchical clustering of the 23 SCPs (derived from the full sample decomposition) based on their pairwise genetic correlations revealed a primary division between the anterior and posterior cerebellum, with additional separations between medial and lateral regions. The full genetic correlation matrix can be found in Supplementary Data 6.

**Fig. 2 | Multivariate GWAS analysis of the 23 cerebellar structural covariance patterns revealed 351 independent genome-wide significant (GWS) loci. a** The upper half of the Miami plot shows the main results from the multivariate analysis. The lower half displays results from a series of 23 univariate analyses (corrected for multiple comparisons using the standard min-P approach), as well as results from a univariate analysis of total cerebellar grey matter (marked in orange). Euler diagrams showing the relative numbers of - and overlaps between - candidate SNPs (in thousands) mapped by the three analysis approaches employed in the current study (**b**) the current and four recent studies reporting genetic associations with cerebellar morphology (**c**) as well as results from multivariate GWASs on hippocampal and cerebrocortical morphology (**d**). For full results on overlap between all cerebellar candidate SNPs and other brain phenotypes, see Supplementary Data 8.

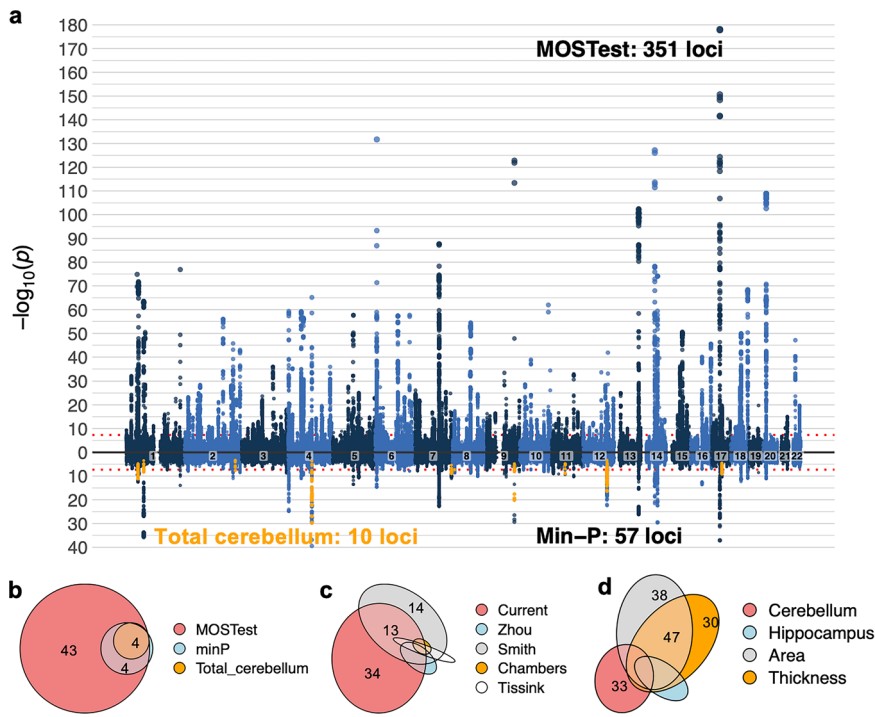

To validate our analysis approach, we computed genetic correlations (using LD-score regression, LDSC[35]) between univariate GWAS results on the comparison features (see Manhattan- and QQ-plots in Supplementary Fig. 5) and previously published neuroimaging GWAS studies on these brain phenotypes. Results showed a mean $r_g$ of. 90 (range. 80–0.99, see Supplementary Data 3). For the 23 cerebellar morphological features, we used the univariate GWAS summary statistics (see Supplementary Figs. 6, 7 for Manhattan- and QQ-plots) to compute genetic correlations between discovery ($n = 27,302$) and replication ($n = 11,264$) samples using LDSC[35]. These genetic correlations were high (mean rG.92; range: 83–1), indicating reliable genetic signals (see Supplementary Data 4).

Genetic complex trait analysis (GCTA[36]) revealed SNP-based heritability estimates ($h^2$) ranging from 0.33 to 0.44 (Fig. 1c and Supplementary Data 5). Analyses of total cerebellar volume ($h^2 = 0.35$), estimated total intracranial volume ($h^2 = 0.41$) and the 9 cerebral comparison phenotypes ($h^2$ range 0.26 to 0.45) gave a similar range of heritability estimates (Supplementary Data 5).

Hierarchical clustering of cerebellar features based on their bivariate genetic correlation matrix (GCTA-based rG ranging from 0.35 to 0.98, Supplementary Data 8) revealed a primary anterior-posterior division running along the horizontal fissure separating Crus I and Crus II, with secondary divisions grouping features into more lateral or medial, as well as more anterior and posterior features within the major anterior and posterior regions (Fig. 1d). Of note, the primary division along the horizontal fissure was also evident from the (genetically naïve) two-component NNMF decomposition, while medial-to-lateral divisions already began to emerge with a model order of three (see Supplementary Fig. 1).

In order to examine whether this phenotypic and genetic correlation structure was also reflected in regional cerebellar gene expression patterns, we used the abagen toolbox[37] to extract Allen Human Brain Atlas[38] gene expression profiles for 22 of the 23 morphological features and computed their bivariate Pearson correlations (across 15,631 genes; Supplementary

Data 7) and hierarchical clustering solution (Supplementary Fig. 8). The anterior-posterior boundary across the horizontal fissure was also evident in the gene expression data, which in addition highlighted distinct gene expression patterns for the posterior midline (grouped together with the horizontal fissure), as well as for the most lateral regions of the cerebellar cortex.

## Multivariate GWAS reveals 351 genetic loci associated with cerebellar morphology

Figure 2a shows the main results for the multivariate GWAS across the 23 cerebellar structural covariance patterns applying MOSTest[17]. We observed 35,098 genome-wide significant (GWS) SNPs, which were mapped by FUMA[39] to a total of 51,803 candidate SNPs by adding reference panel SNPs in high LD ($r > 0.6$) with GWS SNPs. The 51,803 candidate SNPs (see Supplementary Data 8) were represented by 1936 independent significant SNPs and 560 lead SNPs in 351 genomic loci (see Supplementary Data 10). The LDSC intercept of the MOSTest summary statistics was 1.0352 (i.e., indicative of no or minimal genetic inflation), while the QQ-plot of results based on permuted data (under the null hypothesis) confirmed the validity of the MOSTest analytical approach (Supplementary Fig. 9).

Annotation of all candidate SNPs using ANNOVAR[40] as implemented in FUMA[35] revealed that the majority of candidate SNPs were intronic (57.8%) or intergenic (38.3%). While only 0.7% were exonic, about 81% of the candidate SNPs were assigned minimal chromatin states between 1 and 7 (i.e., open chromatin states), implying effects on active transcription[37] (see Supplementary Data 8-9).

We evaluated the robustness of these multivariate results using a multivariate replication procedure established in Loughnan et al.[41], which computes a composite score from the mass-univariate z-statistics (i.e., applying multivariate weights from the discovery sample to the replication sample input data) and then tests for associations between this composite score and genotypes in the replication sample (for mathematical formulation see Loughnan et al.[41]). Results showed that 97% of the 339 loci lead SNPs present in both samples (i.e., 94% of the 351 reported loci lead SNPs from the discovery sample) replicated at a nominal significance threshold of $p < 0.05$ (Supplementary Fig. 10 and Supplementary Data 10), and that 74% remained significant after Bonferroni correction (Supplementary Data 10).

**Fig. 3 | Loci lead SNPs show spatially heterogeneous and replicable effects across the cerebellar cortex.** The 351 loci lead SNPs identified by MOSTest are plotted as a function of main overall effect across all cerebellar features (x-axis: mean Z-score) and most extreme effect for any single cerebellar feature (y-axis: most extreme Z-score across features), and color coded by SNP discovery method. The cerebellar flat-maps show discovery (left) and replication (right) sample regional distributions of Z-scores (color-scale range from −10 to 10) for a few selected lead SNPs (see Supplementary Data 10 for individual feature Z-scores for all 351 discovery sample loci lead SNPs). SNP rs7877685 was only present in the discovery sample.

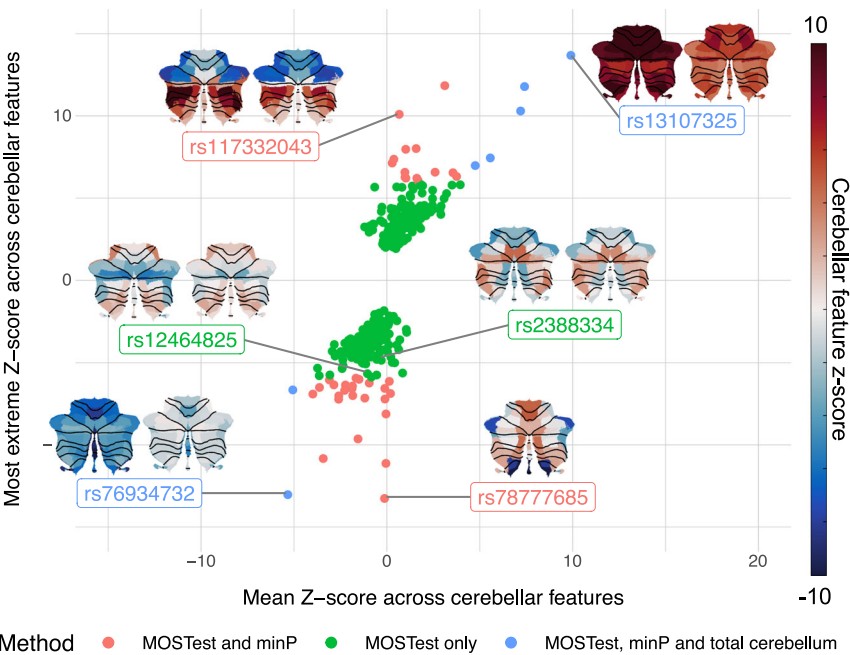

Moreover, 99% of loci lead SNPs showed the same effect direction across discovery and replication samples (Supplementary Fig. 9 and Supplementary Data 10).

In addition, we assessed the robustness of the multivariate patterns by computing bivariate correlations between feature z-score vectors assigned to the discovery sample lead SNPs in an independent multivariate GWAS (MOSTest) performed on the replication sample. These correlations (restricted to the 339 loci lead SNPs present in both samples) were relatively high (mean $r$ 0.70, see Supplementary Data 10 and Supplementary Fig. 11). Figure 3 and Supplementary Fig. 11 also give some examples of discovery and replication sample multivariate patterns projected back onto the cerebellar cortex.

To compare our main multivariate MOSTest results to univariate approaches, the lower part of the Miami plot in Fig. 2a, b, displays results from a set of univariate GWASs on the cerebellar morphological features (which yielded 8370 candidate SNPs and 57 genomic loci, corrected for multiple comparisons using the min-P approach[42,43], see also Supplementary Figs. 6, 7 for univariate Manhattan and QQ-plots), as well as the 4044 candidate SNPs and 10 significant loci resulting from the univariate GWAS on total cerebellar grey matter volume (marked in orange) 0.52 of the 57 loci identified in the univariate analyses of regional cerebellar features overlapped 55 of the 351 loci identified using the multivariate method, while 9 out of 10 loci identified in the univariate analysis of total cerebellar volume were also identified in the multivariate analysis (the slightly mismatching numbers are due to loci of unequal lengths, causing some larger loci to overlap with several smaller loci). Thus, our multivariate analysis of regional cerebellar features increased the locus yield ~35-fold relative to analyzing total cerebellar volume and ~6-fold relative to performing a set of univariate analyses on the same regional features.

We next compared the current findings with previously reported genetic loci for cerebellar morphology by extracting summary statistics from two recent GWAS studies using the UKBB sample ($n$ = 19k[44] and 33k[45]) that included regional cerebellar volumes among the full set of analyzed brain imaging-derived phenotypes (101[44] and 3144[45], respectively), as well as two recent GWASs on total cerebellar volume ($n$ = 33 k[13] and 27 k[14], respectively). Candidate SNPs and independent GWS loci were identified in FUMA using the same settings as for our primary analyses and employing a liberal $p$-value threshold of 5e−8 (i.e., not correcting for the total number of brain imaging features analyzed). Results are displayed in Fig. 2c and Supplementary Data 8 and 10. In brief, we found that 19,527 of the 51,803

candidate SNPs (i.e., 36%) and 123 of the 351 identified genomic loci (i.e., 35%) overlapped with candidate SNPs and loci extracted from these three previous studies. Two additional genetic loci identified in the current study were also reported in a recent study investigating genetic associations with cerebellar shape[46]. Thus, 226 of the 351 (i.e., 64%) genetic loci reported here are novel to the literature on of cerebellar morphology genetics (see Supplementary Data 10.

Overlap of cerebellar candidate SNPs and genetic loci with results from recent multivariate analyses of hippocampal and cerebrocortical morphology are displayed in Fig. 2d and Supplementary Data 8 and 10 (final columns). Of note, we found that 32 and 29 percent of the candidate SNPs discovered here for the cerebellum overlapped with candidate SNPs for vertex-wise cerebrocortical surface area and thickness[18], respectively, while 11.4 percent overlapped with candidate SNPs found for hippocampal subregions[19] 0.95 of the 351 genetic loci overlapped loci linked to the other multivariate brain phenotypes (Supplementary Data 10). Thus, 64% of the candidate SNPs and 73% of genetic loci appeared to be specifically associated with cerebellar morphology.

## Significant genetic variants show heterogeneous effects across the cerebellar cortex, influencing both regional and total volumes

A major advantage of our multivariate analysis approach is its sensitivity to both highly localized and more generally distributed effects of SNPs on cerebellar morphology. This is illustrated in Fig. 3, which displays the 351 loci lead SNPs as a function of both the most extreme individual Z-score across all cerebellar features (e.g, analogous to the strongest "local" effect) and of the mean Z-score across these features (i.e. analogous to the main effect on overall cerebellar volume).

As can be seen, some loci lead SNPs (e.g. rs13107325; rs76934732) show pronounced positive or negative mean z-scores, indicating a relatively consistent direction of effect across cerebellar features. See also inset figures displaying feature Z-scores projected back onto the cerebellar cortex. Many of these SNPs also emerged in the univariate analysis of total cerebellar volume and were recently reported in GWASs on total cerebellar volume[13,14].

Many other loci lead SNPs, however, show strong "local" signals with opposite effect directions across features, yielding very weak global signals (e.g., rs117332043; rs78777685). Thus, while several of the most significant SNPs in this category have previously been reported in GWASs, including local cerebellar morphological features[44,45], they did not emerge from

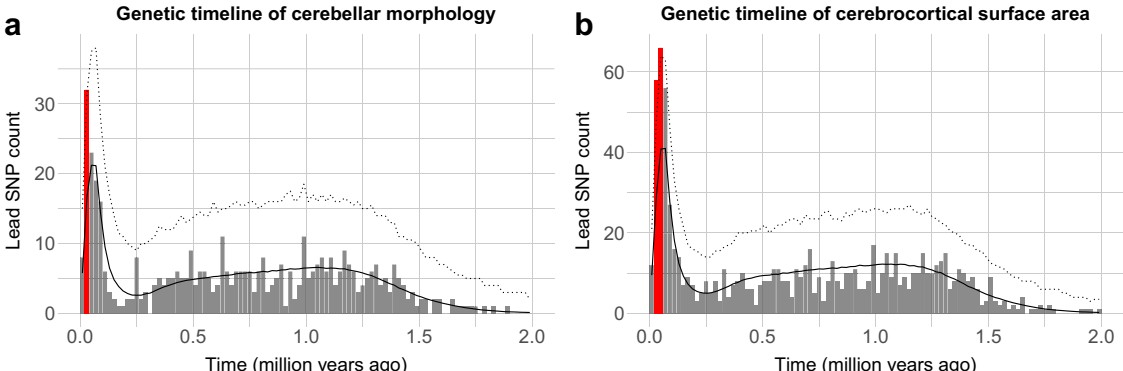

**Fig. 4 | Lead SNPs associated with cerebellar morphology are enriched for evolutionarily recent mutations in the human genome. a** Histogram of estimated SNP age (ranging from 0 to 2 million years, in bins on 20,000 years) for 548 independent lead SNPs associated with cerebellar morphology. The solid black and dotted lines denote the mean and upper 95th confidence interval (Bonferroni corrected across 100 time-bins) derived from a null model constructed from 10,000 of equally sized sets of SNPs randomly drawn from the Human Genome Dating Atlas of Variant Age (after matching these to cerebellar lead SNPs in terms of minor allele frequencies). Red bars denote time-bins of significant positive enrichment. **b** Histogram showing comparative evolutionary enrichment effects for lead SNPs identified in a multivariate GWASs of cerebrocortical area (for cerebrocortical thickness, see Supplementary Fig. 12), See Supplementary Data 11-12 for full numerical results.

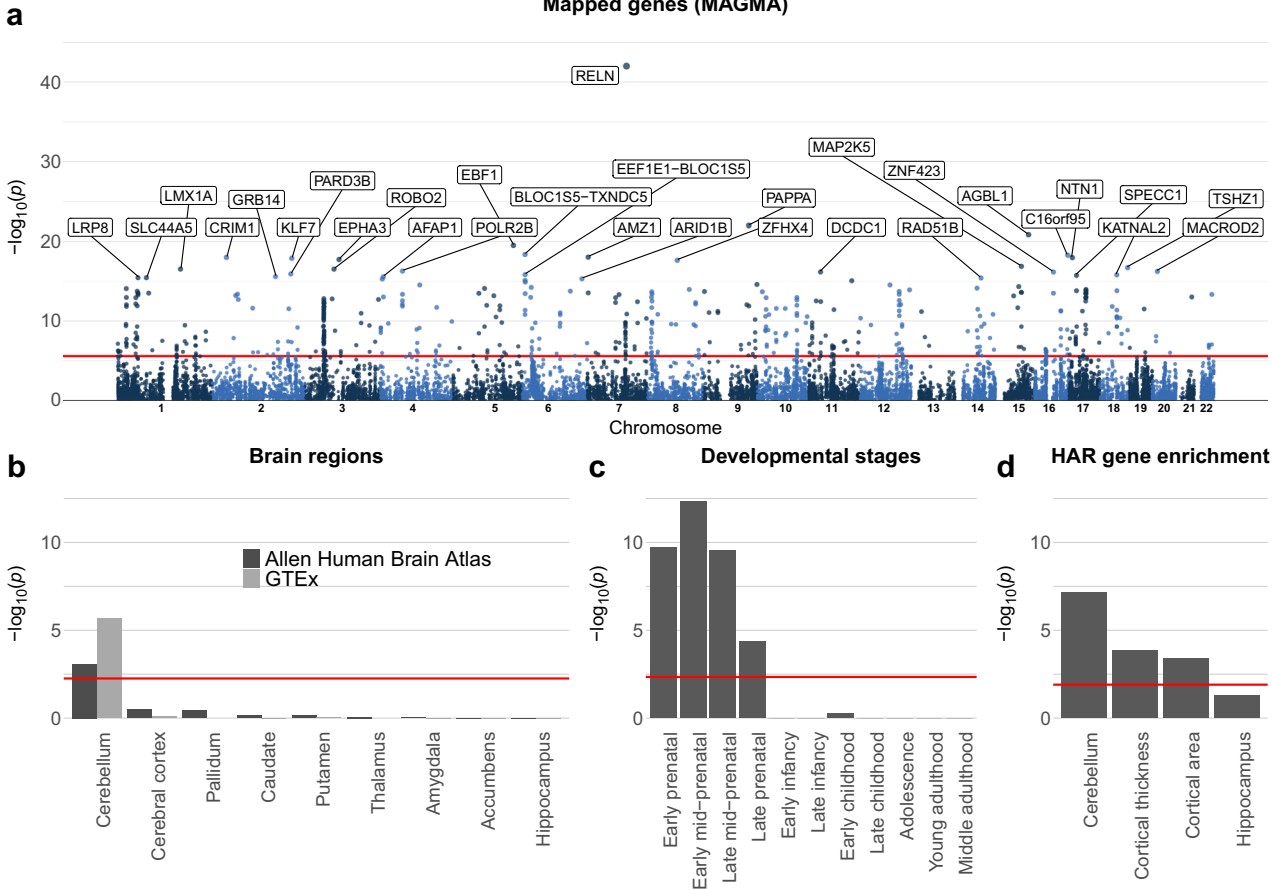

**Fig. 5 | Gene mapping reveals selective enrichment for cerebellar and prenatal brain tissue, as well as for genes linked to human accelerated regions. a** MAGMA mapped the full range of SNPs from the multivariate GWAS to 19.329 protein-coding genes, 534 of which were genome-wide significant (Bonferroni corrected threshold marked with red line), with the 30 most significant genes labeled in the gene-based Manhattan plot. **b, c** MAGMA gene-property analyses revealed selective brain expression in the cerebellum and during prenatal developmental stages.

**d** MAGMA gene-set analysis revealed significant enrichment for sets of genes previously linked to human accelerated regions (HARs). Figure 4d also shows results from comparative analyses using summary statistics from other multivariate GWASs of MRI-based brain, as well as significant results from statistical tests comparing beta-weights for HAR-linked genes across multivariate brain features. Horizontal red lines mark the Bonferroni-corrected significance threshold for each subplot. See Supplementary Data 14–17 for full results.

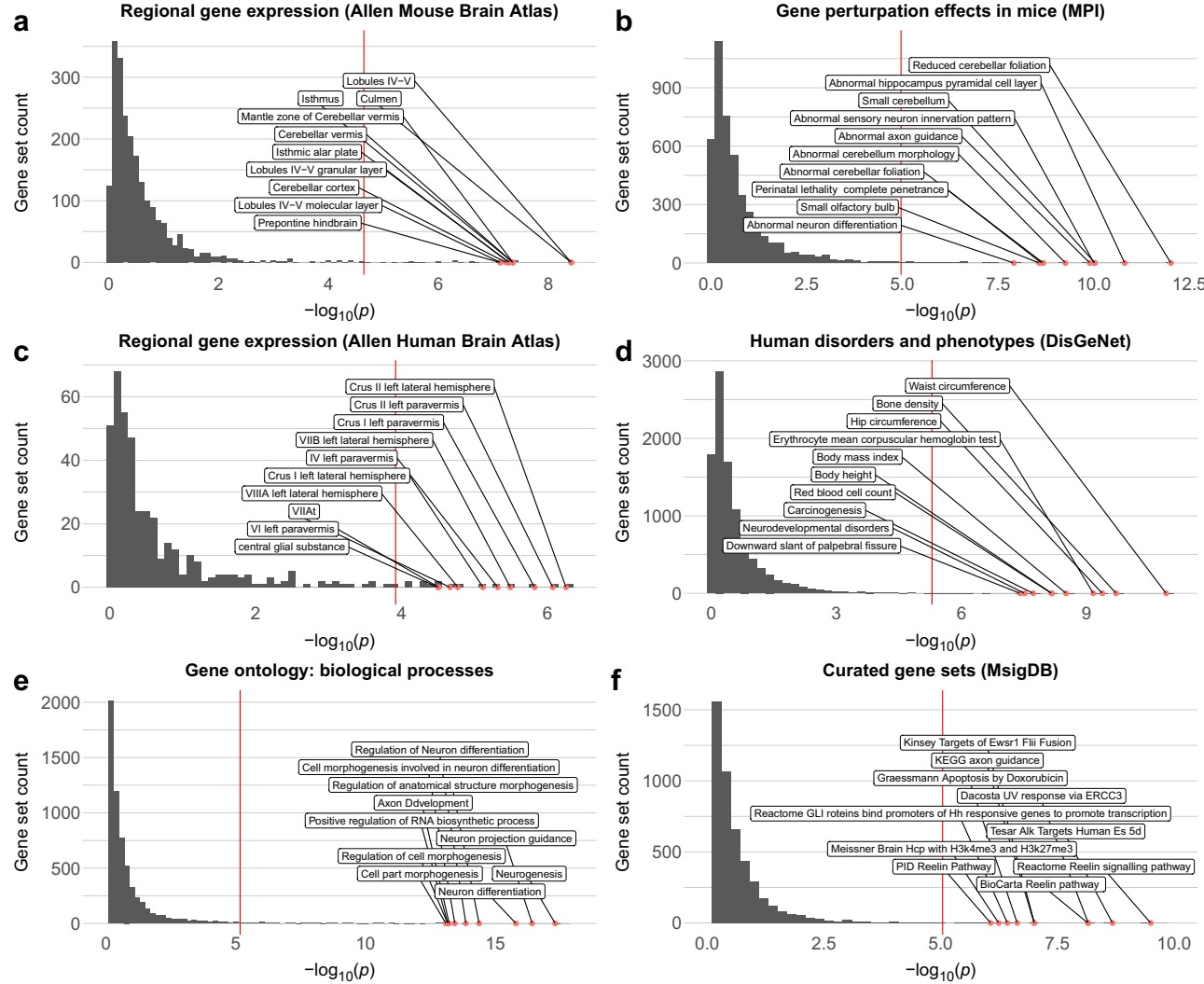

**Fig. 6 | Gene mapping reveals selective enrichment across brain tissues and curated gene sets.** MAGMA gene-set analyses revealed significant enrichment for sets of genes previously linked to preferential expression in human (**a**) and mouse (**c**) cerebellar tissue, as well as effects on cerebellar morphology (and other brain phenotypes) in mouse gene perturbation experiments (**b**) and human clinical disorders and phenotypes (**d**). Significant gene ontology terms were related to neural development (**e**) while curated gene sets highlighted the Reelin signaling pathway (**f**). Across all subplots the x-axis shows the -log10 *p*-value, the y-axis marks the number of gene sets, and the top 10 most significant gene-sets are labelled. Red lines mark the Bonferroni-corrected significance threshold for each subplot. See Supplementary Data 18-23 for full results.

analyses of total cerebellar volume, neither in the current nor in previous studies.

Finally, our multivariate MOSTest approach is also sensitive to SNPs displaying weaker effects distributed across several cerebellar features, often with opposite effect directions (e.g., rs12464825; rs2388334). For this category of SNPs, neither of the univariate methods has sufficient power at the current sample size. In contrast, these two example SNPs robustly emerged from the multivariate analysis (discovery sample *p*-values < 1e−56; replication sample *p*-values < 1e−15).

**Genetic variants associated with cerebellar morphology are enriched for evolutionary recent mutations in the human genome**
We next mapped the evolutionary age of cerebellar lead SNPs (thresholded for linkage disequilibrium at $r^2 < 0.1$) by merging them with a recently published dataset on dated mutations in the human genome[47]. Following the analysis procedure established by Libedinsky et al.[48], we plotted the histogram of dated lead SNPs over the last 2 million years in bins of 20,000 years (Fig. 4a and Supplementary Data 11), and tested for positive or negative enrichment by comparing them to empirical null distributions derived from 10,000 randomly drawn and equally sized sets

of all SNPs in the full human genome dating dataset (after matching them to cerebellum-associated lead SNPs in terms of minor allele frequency; see Online Methods for details).

Results revealed positive enrichment for cerebellar lead SNPs in the time bin ranging from 20-40,000 years ago, i.e., overlapping the Upper Paleolithic, a period characterized by rapid cultural evolution and the first evidence of several uniquely human behaviors (often referred to as behavioral modernity), such as the recording of information onto objects[49].

For comparison, Fig. 4b also shows results from an analysis of lead SNPs identified in a previous multivariate GWAS study of regional cerebrocortical surface area[18], while Supplementary Fig. 12 shows the corresponding histogram for regional cerebrocortical thickness. As can be seen, both these cerebrocortical phenotypes also showed significant enrichment in overlapping time bins (i.e., 20–60,000 years ago). For full numerical results in Supplementary Data 12. Given the relatively low number of independent lead SNPs (range: 548–862 SNPs across phenotypes), we also ran validation analyses using all independent significant SNPs (LD-thesholded at $r^2 < 0.6$, range: 1574–2883 SNPs), which yielded very similar results (see Supplementary Fig. 13 and Supplementary Data 13).

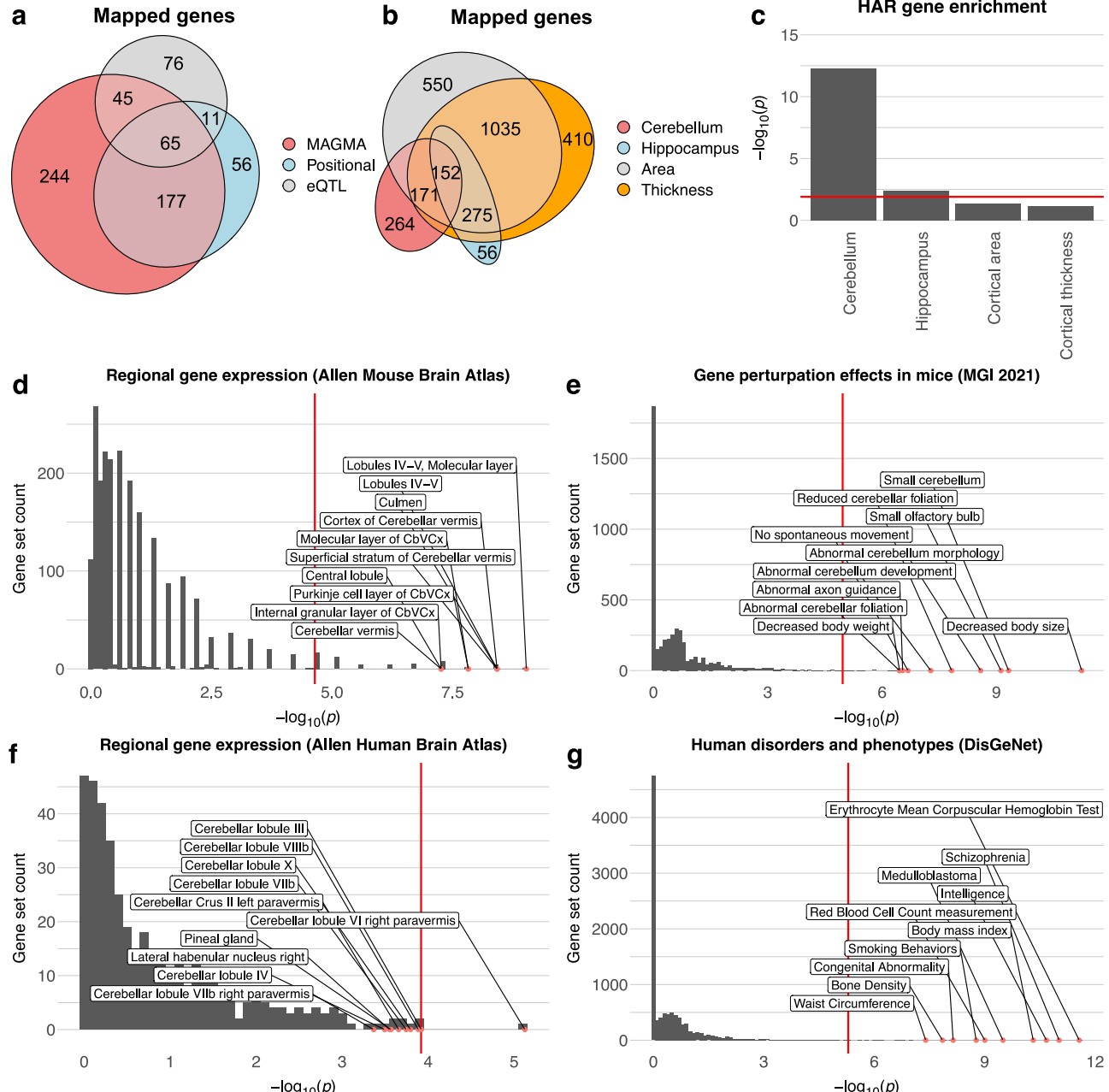

**Fig. 7 | Mapping and functional characterization of plausible causal genes.** The 674 plausible causal genes mapped using three complementary strategies (**a**) show partial overlap with genes mapped to hippocampal[19] and cerebrocortical[18] morphology (**b**); significant enrichment for sets of genes linked to human accelerated regions (HAR) (**c**); selective expression in the cerebellum in mice (**d**) and humans (**e**) as well as effects on cerebellar morphology in mouse gene perturbation studies (**f**) and human disorders and anthropometric phenotypes (**g**). Across all subplots, the x-axis shows the -log10 p-value, the y-axis marks the number of gene sets, and the top 10 most significant gene-sets are labelled. Red lines mark the Bonferroni-corrected significance threshold for each subplot. See Supplementary Data 24–29 for full results.

## Genes associated with cerebellar morphology show selective expression in cerebellar and prenatal brain tissue, as well as enrichment for genes linked to human accelerated regions

To functionally characterize the multivariate GWAS signal, we mapped the full set of GWAS p-values to 19,329 protein coding genes using MAGMA[50] and used the resulting gene-level p-values to test for (1) GWS genes, (2) gene expression in brain tissue; and (3) enrichment for genes linked to human accelerated regions (HARs), i.e. sections of DNA that have remained relatively conserved throughout mammalian evolution, before being subject to a burst of changes in humans since divergence of humans from chimpanzees[51,52]. These analyses yielded a total of 534 GWS genes (i.e., 2.78% of all protein-coding genes, see Fig. 5a and Supplementary Data 14).

Using the full set of 19,329 gene-level p-values in MAGMA gene property analyses revealed significant and specific gene expression in cerebellar and prenatal brain tissue (Fig. 5b, c and Supplementary Data 15, 16), with the selective cerebellar expression seen in two independent datasets (Allen Human Brain Atlas[38] and The Genotype-Tissue Expression (GTEx) Project).

The MAGMA gene set analysis of HAR-linked genes revealed significant enrichment ($p = 7.09e{-}08$) for genes associated with cerebellar morphology (Fig. 5d, Supplementary Data 17). Of note, running this same HAR gene set analysis on summary statistics from recent multivariate GWAS studies on cerebrocortical[18] or hippocampal[19] morphological features yielded similar (for cortical features) or significantly weaker

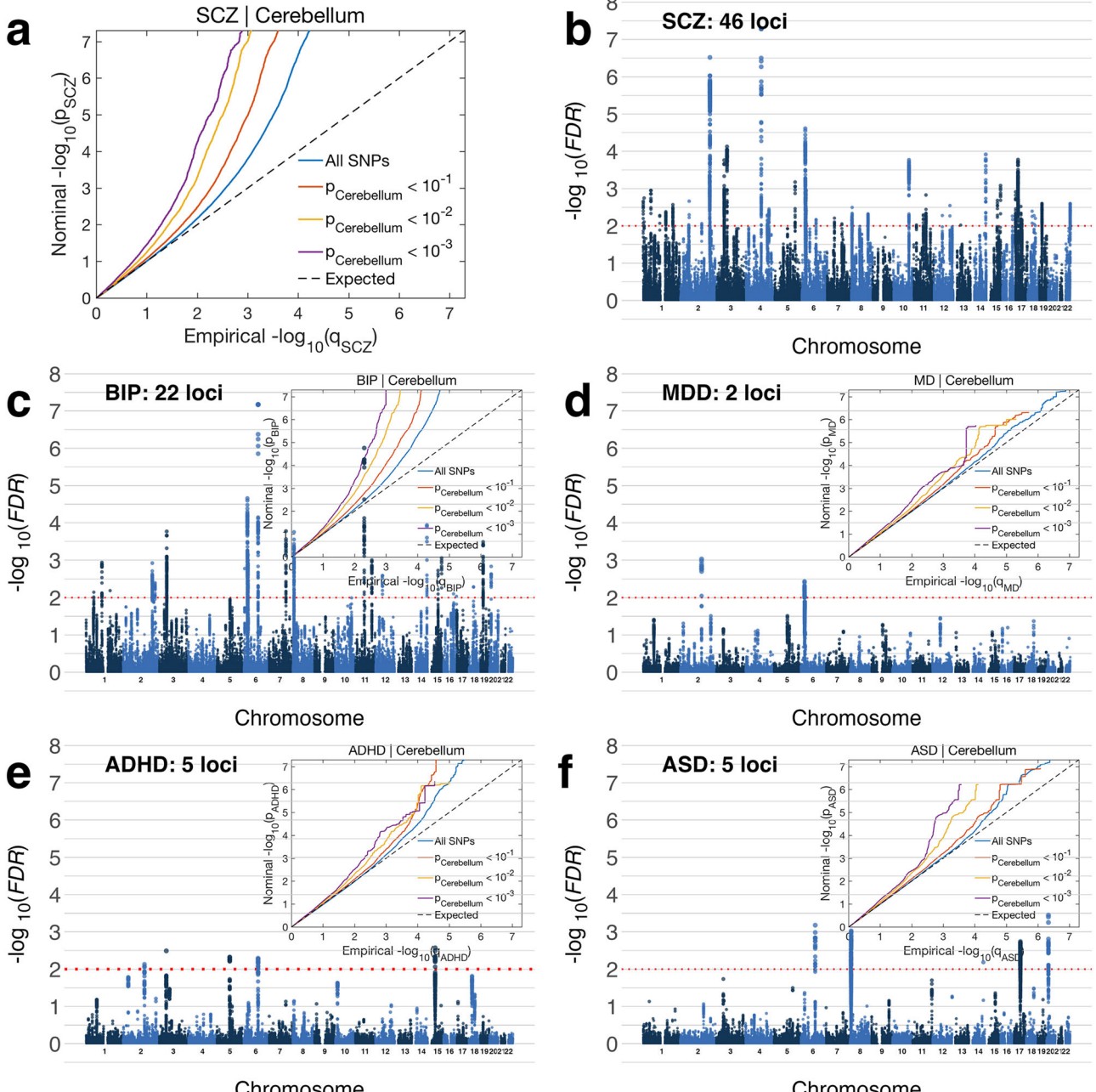

**Fig. 8 | Genetic variants influencing cerebellar morphology overlap with variants associated with five major mental disorders.** Conditional QQ plots (**a** and figure insets in **c**–**f**) show an incremental incidence of association with five mental disorders (leftward deflection) as a function of the significance of association with cerebellar morphology. Manhattan plots (**b**–**f**) show SNPs with significant association with both traits, thresholded at a conjunctional FDR threshold of $p > 0.01$ (red dotted line). SCZ Schizophrenia, BIP Bipolar Disorder, MDD Major Depressive Disorder, ADHD Attention Deficit Hyperactivity Disorder, ASD Autism Spectrum Disorders. See Supplementary Data 30–34 for full results.

(hippocampal features) enrichment effects (Fig. 5d, see Online Methods for processing pipeline).

**Genes linked to human cerebellar morphology show enrichment for gene sets linked to selective cerebellar gene expression, altered cerebellar morphology in mouse models, human clinical/ anthropometric traits, as well as specific biological processes and pathways**

In line with the continuous brain tissue gene expression results described above, we also observed significant and relatively selective enrichment for smaller curated sets of genes previously found to be highly and selectively expressed in mouse (Fig. 6a, Supplementary Data 18) and human (Fig. 6c,

Supplementary Data 20) cerebellar brain tissue, as well as for sets of genes previously shown to affect cerebellar morphology in mouse gene perturbation experiments (Fig. 6b, Supplementary Data 19) and various clinical conditions and anthropometric traits in humans (Fig. 6d, Supplementary Data 21). The most significant gene ontology and curated gene sets from the MSigDB[53,54] database were related to brain development (e.g., neurogenesis, axon guidance, and neuron differentiation, Fig. 6e, Supplementary Data 22), and highlighted the reelin signaling pathway (Fig. 6f, Supplementary Data 23).

As can be seen in Fig. 5a (and Supplementary Data 14), *RELN* (encoding the protein Reelin) was also the most significant gene mapped by MAGMA. See also Supplementary Fig. 14 for a regional locus plot showing the 12 lead SNPs mapped to *RELN* and their associated z-score maps.

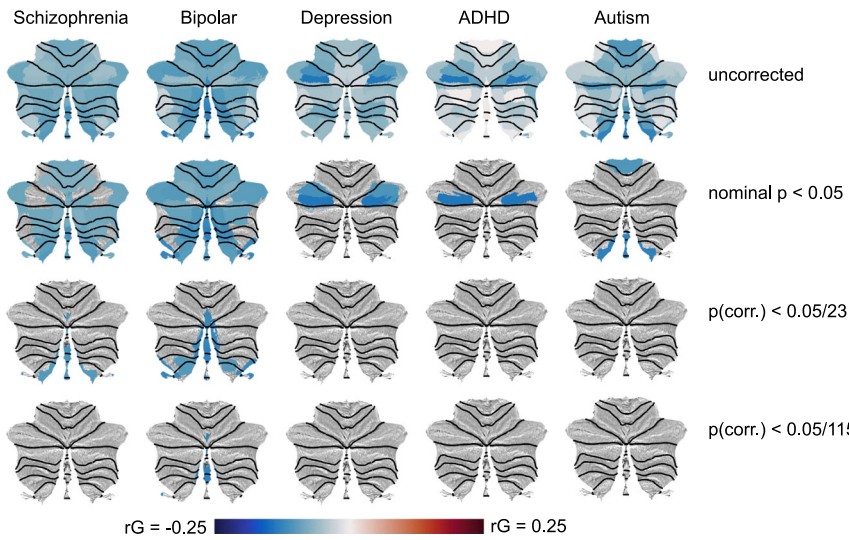

**Fig. 9 | Univariate genetic correlations between cerebellar morphological features and five major mental disorders are negative.** The top row display unthresholded genetic correlations, while these are filtered at increasingly strict statistical thresholds in the following rows, i.e. nominal p-value < 0.05 (second row); Bonferroni correction for the 23 cerebellar features tested (third row); and Bonferroni correction for both 23 features and 5 clinical conditions (bottom row). Black dotted lines denote lobular boundaries. See Supplementary Data 36 for full results.

## Gene mapping reveals sets of plausible causal genes

In addition to the gene-based mapping strategy using all SNPs described above (MAGMA), we also mapped candidate SNPs to genes using two complementary gene mapping strategies: 1) positional mapping of deleterious SNPs (defined as having a CADD-score[55] > 12.37); and 2) mapping of SNPs previously shown to affect gene expression in cerebellar tissues (i.e., eQTL mapping). Across all three strategies, we mapped a total of 674 unique genes; 531 using MAGMA, 310 using positional and 197 using eQTL mapping (see Fig. 7a and Supplementary Data 24). 298 genes were identified by at least two strategies, while 65 genes were mapped by all three strategies. Out of these 674 genes, 61 have previously been associated with cerebellar pathology in humans and/or altered cerebellar morphology in mouse gene perturbation experiments, while 121 have been linked to human accelerated regions (Supplementary Data 24). As can be seen in Fig. 7b and Supplementary Data 24, the 674 genes mapped to cerebellar morphology showed some overlap with genes mapped to hippocampal[19] and cerebrocortical[18] morphology using the same mapping strategies, but 264 genes (39%) appeared relatively specific to the cerebellum.

Results from gene set analyses on the 674 mapped genes (Fig. 7, Supplementary Data 25–29) largely mirrored results from the MAGMA analyses described above, but in addition revealed that this set of 674 mapped genes was also enriched for gene sets associated with several complex clinical phenotypes and anthropometric traits in humans, including cognitive ability, neuroticism and schizophrenia (Fig. 7f, Supplementary Data 29).

Of note, while the full set of mapped genes showed significant enrichment for sets of genes known to alter cerebellar morphology in mouse mutation or knock-down experiments (Supplementary Data 27), we also note that 613 of the 674 mapped genes have not to our knowledge previously been linked to cerebellar development, anatomy or pathology in mice or humans (Supplementary Data 24), and thus constitute potential targets for future gene perturbation experiments in animal models.

Restricting the above analyses to the 298 genes mapped across at least two strategies did not markedly affect the results (Supplementary Data 25-29, final columns).

## Cerebellar morphology shows significant genetic overlap with psychiatric disorders

We finally tested for overlap between the multivariate genetic profile for cerebellar morphology and genetic profiles for five major developmental/psychiatric disorders (attention deficit hyperactivity disorder: ADHD; autism spectrum disorder: ASD; bipolar disorder: BIP; major depressive disorder: MDD; and schizophrenia: SCZ) using conditional/conjunctional FDR analysis[56]. As can be seen in the conditional QQ-plots in Fig. 8, these analyses revealed clear patterns of enriched association with the clinical phenotypes when selecting subsets of SNPs with increasingly stronger association with cerebellar morphology (Fig. 8a and figure insets in (c–f).

See also Supplementary Fig. 15 for QQ-plots depicting the reverse association, i.e., enriched association with cerebellar morphology when conditioning on the association with developmental/psychiatric disorders. Specific genetic variants jointly influencing the two phenotypes were identified using conjunctional FDR analyses at a conservative statistical threshold of p < 0.01. Results revealed shared genetic loci with all disorders; namely, 48 with SCZ, 22 with BIP, 2 with MDD, 5 with ADHD and 5 with ASD (Fig. 6; Supplementary Data 30–34). We mapped lead and candidate SNPs for each of these loci to genes using positional and eQTL mapping and checked for gene overlap across disorders (Supplementary Data 35). Of note, the *LRP8* gene (a HAR-linked gene[51,52] encoding a Reelin receptor) emerged from the conjunctional FDR analyses of both BIP and SCZ, thus again highlighting the Reelin signaling pathway.

To complement these multivariate analyses, we also conducted a set of univariate analyses, using LD-score regression[35] to calculate the genetic correlations between each individual cerebellar feature and the five developmental/psychiatric disorders.

As can be seen in the first row of Fig. 9 (and Supplementary Data 36), genetic correlations with cerebellar morphological features were predominantly negative across diagnoses, indicating that genetic variants associated with a clinical diagnosis tended to also be associated with reduced cerebellar volumes (with 42 out of 115 tested associations showing nominally significant negative correlations; Fig. 9, second row). However, all univariate genetic correlations were relatively weak, and only a few negative genetic correlations with BIP and SCZ survived Bonferroni correction for the 23 features tested. When also correcting for the five clinical conditions, only the negative correlation between BIP and cerebellar feature 23 (primarily overlapping vermal lobules VIIIa and VIIIb) remained significant.

Supplementary Data 37 show genetic correlations between the five disorders and the ten comparison brain phenotypes, as well as total cerebellar volume. In general agreement with the observed pattern for regional cerebellar features, total cerebellar grey matter volume showed nominally significant negative genetic correlations with BIP (rg:- 0.10; p = 0.0052) and SCZ (rg:- 0.09; p = 0.0133). Pallidal volume also showed nominally significant negative genetic correlations with these two disorders (BIP: rg:-0.10; p = 0.0087; SCZ: rg:- 0.08; p = 0.0172, while ADHD displayed negative genetic correlations with estimated total intracranial volume (rg:- 0.15; p = 0.0003) and total cortical surface area (rg:- 0.14; p = 0.0065), as well as a positive genetic correlation with hippocampal volume (rg:- 13; p = 0.0164). Finally, MDD showed nominally significant negative genetic correlations with volumes of the hippocampus (rg:- 0.08: p = 0.025) and thalamus (rg:- 0.09; p = 0.0258). Only the negative genetic correlation between ADHD and

estimated total intracranial volume survived Bonferroni correction across the 55 tests performed.

## Discussion

The current study identified novel features of the genetic architecture of cerebellar morphology, supported the notion of recent changes over human evolution, implicated specific neurobiological pathways, and demonstrated genetic overlap with major mental disorders.

With respect to the main features of cerebellar morphology, it is worth noting that results from our data-driven decomposition of cerebellar grey matter maps (which was not informed by genetic data), the genetic correlation analyses of the chosen 23-feature solution and gene expression data from the Allen Human Brain Atlas all converge on a similar general pattern. The first boundary to emerge in the data-driven decompositions ran along the horizontal fissure separating Crus I and II of lobule VII, reflecting its centrality in characterizing phenotypic variability in cerebellar morphology at the population level. Of note, this boundary also emerged from clustering of the 23 cerebellar morphological features based on their bivariate genetic correlations or gene expression profiles (based on the Allen Human Brain Atlas). This latter finding essentially mirrors results from a recent report using a different analysis strategy on the same gene expression data[57]. Interestingly, the horizontal fissure has been suggested to mark the border between two separate cerebellar representations of the cerebral cortex, with a possible third representation in lobules IX-X[58]. The current cerebellar results thus complement previous work on the hierarchical genetic organization of the cerebral cortex, which has identified the Rolandic fissure (separating the frontal and parietal lobes) as a main boundary with respect to genetic effects of effects of surface area[59], as well as a superior-inferior gradient for genetic influences on cortical thickness[60].

Our multivariate GWAS using MOSTest identified 351 independent GWS loci associated with cerebellar morphology, increasing the yield ~35-fold relative to analyzing total cerebellar volume and ~6-fold relative to performing a set of univariate analyses on the same regional features in our current sample. 329 (94%) loci from the multivariate analyses were replicated in an independent sample, indicating robust results. After applying a liberal threshold to summary statistics from previous well-powered studies, we find that 228 (65%) of our reported loci are novel. Importantly, among the genes mapped to these novel loci we find several that are known to play important roles in cerebellar development in mice (e.g., *RORA*[61], *FGF8*[62] and *BAHRL1*[63], see Supplementary Data 23). While candidate SNPs associated with cerebellar morphology partially overlap with SNPs previously mapped to other multivariate brain phenotypes, we note that a substantial number of SNPs appear to be relatively selectively linked to cerebellar morphology, a finding that is in in line with the distinct gene expression profile found for the cerebellum[38].

SNP- and gene-level results from the current study also bolster—and refine—the notion of relatively recent changes in cerebellar morphology over human evolution[2,4,15,16]. Specifically, we found that lead SNPs associated with cerebellar morphology were enriched for SNPs with an estimated age of 20–40 thousand years. This overlaps the Upper Paleolithic (10–50k years ago), a period characterized by rapid cultural evolution, and coinciding with the first evidence of several uniquely human behaviors (often referred to as behavioral modernity), such as the recording of information onto objects[49]. Converging evidence for changes in brain anatomy around this evolutionary period comes from the fossil scull record, where only fossils dated less than 35.000 years old fall within the range of shape-variation seen in modern humans[64,65]. Of note, one key feature of modern skulls is their "globular" shape, in part characterized by an enlarged posterior fossa (the cranial compartment housing the cerebellum)[64,65]. A similar pattern of enrichment for recent evolutionary time bins (20–60 thousand years ago) was also found when analyzing lead SNPs derived from previous studies on multivariate cerebro-cortical[18] morphology.

Gene level analyses further showed that genes associated with inter-individual variation in cerebellar morphology are enriched for genes linked to human accelerated regions (HARs)[51] of the genome. HARs denote previously conserved regions of the genome that were subject to a burst of changes in humans after the divergence of humans from chimpanzees about 6–8 million years ago[51]. Of note, a recent GWAS on total cerebellar volume found no enrichment for HAR-linked genes[14], suggesting that SNPs associated with regional cerebellar variation may be driving this effect. When comparing this finding with results from other multivariate brain phenotypes, we observed that HAR gene enrichment was nominally stronger for cerebellar morphology than for vertex-wise cerebrocortical thickness and area, and significantly stronger than for hippocampal regional volumes (Fig. 5d).

Together, these SNP- and gene-level results suggest that genetic variants influencing cerebellar morphology in modern humans have been subject to selection over relatively recent human evolution, and that cerebellar changes—in concert with other brain regions—may thus have played a central part in the emergence of uniquely human cognitive abilities.

Results from the MAGMA gene property and gene set analyses bolster our confidence in the genetic signal by showing selective gene expression in human cerebellar brain tissue across two independent datasets (Allen Human Brain Atlas and GTEx.8) and significant enrichment for sets of genes that have previously been shown to affect cerebellar morphology in mouse gene perturbation experiments. The MAGMA results further show significant enrichment for sets of genes known to play key roles in neurodevelopment (e.g., neurogenesis & axon guidance) and preferential expression in prenatal brain tissue, thus supporting a primarily developmental origin of genetically determined effects on adult cerebellar morphology. Of note, our MAGMA enrichment analysis of curated gene sets strongly implicated the Reelin signaling pathway. Indeed, both the gene coding for the Reelin protein (*RELN*) and genes coding for its two receptors (*LRP8* & *VLDLR*) were identified across at least two gene mapping strategies, with *RELN* emerging as the single most significant gene by MAGMA. The Reelin pathway is known to play important roles in neurodevelopment (e.g. neuronal migration), and mutations in the *RELN* and VLDLR[66] (and to a lesser extent LRP8; also known as ApoER2[67]) have been associated with cerebellar malformations and/or dysfunction. LRP8 is also among the genes linked to human accelerated regions (HARs) of the genome.

The sets of genes mapped by the three complementary mapping strategies provide a database for future studies investigating the genetic architecture of cerebellar morphology. For instance, we mapped 616 genes associated with inter-individual variability in human cerebellar morphology that have not yet to our knowledge been examined in mouse gene perturbation experiments and/or associated with cerebellar pathology in humans. Among these, we highlight *MAP2K5* and *GRB14*, two HAR-linked genes mapped across all strategies and associated with lead SNP *p*-values < 1e−50, but whose functions in the brain are largely unknown.

The reported results for previously discovered variants, loci and genes add important information regarding regional effects on cerebellar morphology. For instance, while genetic variants linked to the *RELN* gene have previously been associated with volumes of cerebellar vermal lobules VI-X and hemispheric lobule IX[44,45], we here mapped 12 lead SNPs to *RELN* showing heterogeneous effects across the entire cerebellar cortex (but with peak effects overlapping previously described midline and posterior cerebellar regions, see Supplementary Fig. 13).

The observed genetic overlap between cerebellar morphology and the five mental disorders reinforces the recent notion of the cerebellum as a key brain structure in complex clinical traits and disorders[6–10]. Across the five diagnoses, the strongest evidence for genetic overlap with cerebellar morphology was found for SCZ and BIP, likely at least in part because these disorder GWASs were relatively well-powered. While an in-depth discussion of genetic loci jointly influencing psychiatric disorders and cerebellar morphology is beyond the scope of this report, we note that the Reelin pathway again emerges in the genetic overlap analyses for SCZ and BIP. Specifically, the current finding of *LRP8* (a reelin receptor, and HAR-linked gene) as one of the genes jointly associated with cerebellar morphology and the aforementioned severe mental disorders points towards a potential molecular pathway involved in the cerebellar abnormalities previously

reported in SCZ[10,68,69] and BIP[68,69]. Indeed, in line with its key importance for brain development and function, the Reelin pathway is also increasingly seen as relevant for a wide range of neurodevelopmental, psychiatric and neurodegenerative disorders[70]. Of particular relevance to the current findings, converging evidence supports LRP8 as a key susceptibility gene for psychosis[71].

The main limitations of the current study concern the ancestral homogeneity of the sample, the sample size, the exclusion of very rare genetic variants and the limitations on follow-up analyses placed by multivariate (relative to univariate) test-statistics. Limiting the sample to participants of European ancestry was deemed necessary considering the current state of the multivariate GWAS methods used but may limit the generalizability of our findings. Second, while the current sample size is large in comparison with previous imaging genetics studies, it is still relatively small in comparison to GWASs of other complex human phenotypes (e.g., intelligence, with a current n of > 3 million[72]). Moreover, very rare genetic variants (MAF < 0.005) were excluded from the current multivariate GWAS, but are likely to include a number of variants with relatively large effect sizes on complex human traits[73]. Thus, future studies using larger and more diverse samples – as well as whole exome and/or genome sequencing – are likely to discover more of the genetic variants associated with cerebellar structure. Finally, while multivariate GWAS increases the power to detect genetic variants associated with brain phenotypes relative to univariate approaches, it also places important limitations on the possible follow-up analyses (e.g., genetic correlations or Mendelian randomization), which require directional effects. Although comprehensive follow-up analyses of univariate GWAS results fall beyond the scope of the current report, we have made all summary statistics (univariate and multivariate) publicly available for the research community.

In conclusion, the current results enhance our understanding of the genetic architecture of human cerebellar morphology, provide supporting evidence for cerebellar morphological changes during the last ~6–8 million years of human evolution, and reinforce the notion of cerebellar involvement in several mental disorders by demonstrating significant genetic overlap.

## Materials and methods
### Participants
For our main analyses, T1-weighted MR images, demographic and genetic data from 39,178 UK Biobank participants were accessed using access number 27,412. All participants provided informed consent and the UK Biobank was approved by the National Health Service National Research Ethics Service (ref11./NW/0382). All ethical regulations relevant to human research participants were followed. After removing 1043 participants who either had missing genetic data or had withdrawn consent (as of 19.11.2019), data from 38,135 participants remained for the main analysis (age range: 44.6–82.1; mean age: 64.1, 51.9% female). Following quality control procedures (QC, see below), 28,212 UK Biobank participants of European descent remained for the main analyses (age range: 45.1–82.1; mean age: 64.1, 55.1% female), 910 of which were identified as close relatives and removed prior to genetic analyses (see below), leaving a final sample for the primary analyses of 27,302 (age range: 45.1–82.1; mean age: 64.1, 54.9% female). For the replication sample, we accessed a newer release of UK Biobank participants ($n = 48,045$) and removed the 27,302 participants included in the primary analyses. After running through identical QC procedures as for the main sample (although applied only to the cerebellar features of primary interest and covariates included when analyzing these), the replication sample consisted of 11,264 unrelated UK Biobank participants of European descent (age range: 46.1–83.7; mean age: 66.8, 46.9% female).

### Initial MR image processing
MRI data was first processed using the recon-all pipeline in Freesurfer 5.3[74], yielding a large number of brain features. Of these, we retained measures of estimated total intracranial volume (eTIV), total cerebro-cortical surface area, average cerebro-cortical thickness as well as the volumes of seven subcortical structures (hippocampus, amygdala, thalamus, pallidum, putamen, caudate nucleus and nucleus accumbens) as cerebral (or global, in the case of eTICV) comparison regions for our main cerebellar analyses. These anatomical features were averaged across hemispheres, yielding a total of ten comparison phenotypes.

Next, the bias-field corrected T1-images from the FreeSurfer analyses were analyzed using the cerebellum-optimized SUIT-toolbox[75]. In brief, SUIT isolates the cerebellum and brain stem from T1-images, segments cropped images into grey and white matter, and normalizes these tissue probability maps to a cerebellum-specific anatomical template. After multiplying the grey matter maps with the Jacobian of the transformation matrix (i.e. preserving information about absolute volumes), we extracted grey matter intensity values overlapping the 28 cerebellar lobular labels in the probabilistic SUIT-atlas. Since cerebellar volumetric indices showed high correlations across the two hemispheres (range 0.87–0.96; mean 0.94), we created mean volumetric measures by averaging across hemispheres. Finally, we also combined the two smallest regions in the SUIT-atlas (Vermis Crus I and II located in the midline, average volumes: 2.9 and 293.4 mm$^2$, respectively) to create a new Vermis Crus region (average volume: 296.2 mm$^2$). This procedure reduced the 28 cerebellar lobular volumes to 16 morphological indices. Total cerebellar grey matter volume was defined as the sum of all 28 lobular labels in the SUIT-atlas.

### Quality control procedures
After excluding 4818 UKBB participants of non-European origin, anatomical indices from the remaining 33,317 participants went through an iterative quality control (QC) procedure. First, we excluded 639 subjects with a mean Euler number below −217, indicating poor MRI quality[34], as well as 12 subjects with missing and 90 subjects with zero values for any of the key cerebellar or cerebral brain measures of interest. Next, we used general additive models (GAM, implemented in the R-package "mgcv") in order to model the effects of age (estimated as smooth functions for males and females separately, using cubic splines with 10 knots), sex, and scanner site on estimated total intracranial volume (eTIV) and mean thickness of the cerebral cortex. Specifically, we used the following formula:

*GAM_model < -gam(Anatomical feature ~ s(Age, bs = "cs", by = Sex, k = 10) + as.factor(Sex) + as.factor(Scanner), data = UKBB_cerebellum_GWAS, select = TRUE, method = "REML")*

Adjusted eTIV and mean cortical thickness indices were then created by reconstructing the data using the intercept and residuals from this model (i.e., removing effects of age, sex and scanner), before identifying and rejecting potential outliers, defined as +/− 3 median absolute deviations (MAD)[76] from the median of these adjusted values. Data from 350 subjects were rejected based on these criteria.

For the remaining cerebral anatomical measures, as well as total cerebellar volume, this procedure was then repeated, with scaled eTIV as an additional predictor. Since previous studies have demonstrated that the relationship between regional brain volumes and intracranial volume is not strictly allometric[77,78], we estimated the effect of eTIV using cubic splines with 10 knots. Specifically, we used the following formula:

*GAM_model < -gam(Anatomical feature ~ s(Age, bs = "cs", by = Sex, k = 10) + as.factor(Sex) + as.factor(Scanner) + s(eTIV_scaled, bs = "cs", k = 10), data = UKBB_cerebellum_GWAS, select = TRUE, method = "REML")*

In order to be maximally sensitive to outliers in relative cerebellar volumes, we replaced eTIV with total cerebellar volume in the GAM models of cerebellar regions of interest (i.e., SUIT atlas regions), using the following formula:

*GAM_model < -gam(Anatomical feature ~ s(Age, bs = "cs", by = Sex, k = 10) + as.factor(Sex) + as.factor(Scanner) + total cerebellar volume, data = UKBB_cerebellum_GWAS, select = TRUE, method = "REML")*

Adjusted cerebral and cerebellar indices were then created by reconstructing the data using the intercept and residuals from these models (i.e., removing estimated effects of age, sex and eTIV or total cerebellar volume),

before rejecting 1792 participants with potential outlier cerebral indices and 2222 participants with potential outlier cerebellar indices (MAD > +/− 3).

This iterative QC procedure resulted in the rejection of 5105 (i.e., 15.3%) of the original 33,317 datasets, leaving 28,212 datasets for further analysis.

## Non-negative matrix factorization

Cerebellar grey matter maps from 28,212 subjects passing the iterative QC procedure were smoothed with a 4 mm full-with-half-maximum gaussian kernel in SPM12[79], concatenated across all subjects and multiplied with a binary mask to isolate voxels located in the cerebellar cortex. This cerebellar cortical mask was constructed by multiplying a binary mask containing all 28 cerebellar lobules of the SUIT-atlas with the thresholded (at a value of 0.1) mean (unsmoothed) grey matter segmentation across all 28,212 participants.

The smoothed, concatenated and masked grey matter maps were then subjected to orthogonal projective non-negative matrix-factorization (OPNMF)[29], in order to derive data-driven parcellations of regional cerebellar grey matter volume.

Non-negative matrix factorization (NNMF) is a blind source separation technique that allows structural brain networks to be described in a hypothesis-free, data-driven way by identifying patterns of covariation in the data. In contrast to alternative techniques, such as principal component analysis and independent component analysis, which yield components with both positive and negative weights that are often difficult to interpret, NNMF produces a sparse, positive-only, parts-based representation of the data. Importantly, NNMF has previously been proven effective in estimating covariance patterns in neuroimaging data while providing an easier interpretation of the results than other matrix decomposition techniques such as principal component analysis (PCA) or independent component analysis (ICA)[29–33].

Briefly, NNMF decomposes an input matrix (voxels×subjects) into two matrices; a component matrix W (voxels×k) and a weight matrix H (k×subjects) where $k$ is the number of components that needs to be specified by the user. Here, we used an implementation of orthogolnal projective non-negative matrix factorization previously used in a number of publications[29–33] and downloaded from https://github.com/asotiras/brainparts. Given the large number of participants, we used the opnmf_-mem.m function, which has been optimized for high-dimensional data. The function was run with default parameter settings, exept for maximal number of iterations, which was increased to 200k, in order to ensure full convergence across all tested model orders.

## Model order selection

Since the resulting parcellations are highly dependent on the requested number of components (or model order) specified, we tested model orders ranging from 2 to 30 in steps of 1, as well as from 30 to 100 in steps of 10. Binarized winner-winner-takes-all (i.e. assigning each voxel to the NNMF component with the highest loading) maps of the resulting decompositions projected onto a flattened representation of the cerebellar cortex are shown in Supplementary Fig. 1.

Resulting NNMF decompositions were then evaluated based on two criteria; 1) how much of the variance in the original input data a given NNMF solution (i.e., component maps and subject weights) could explain and on 2) how reproducible the component maps were. As a metric of change in explained variance we used the change of the Frobenius norm of the reconstruction error. With increasing model orders the variance explained will always increase and the reconstruction error decrease, but if the decrease in the reconstruction error (or gradient) levels off, this indicates that the intrinsic dimensionality of the data might have been approximated (and that the subsequent increase in explained variance can largely be attributed to fitting random noise in the input data). In order to assess reproducibility, we split the full dataset into two equal sets (matched with respect to scanner site, n = 14,105 and 14,107, respectively), and ran NNMF on each split-half sample. For each set of independent NNMF parcellations,

we computed two reproducibility indices. First, for each model order we matched components across split-half runs using the Hungarian algorithm[80], before computing the spatial correlations between matched component maps, and extracting the median correlation across all matched components as our first reproducibility index. For our second reproducibility index, we first computed one overall—and categorical—component map using a "winner-takes-all"-approach, i.e., assigning each voxel to the NNMF component with the highest loading. Next, we calculated the adjusted Rand index (ranging from 0 to 1, with higher values indicating greater similarity[81], across the two categorical parcellation maps for each model order as our second metric of reproducibility.

As can be seen in Supplementary Fig. 2a, for lower model orders, increasing the model order resulted in a sharp decrease in the reconstruction error, indicating that models with more components resulted in a significantly better fit to the data. However, after reaching model orders between 15 and 30, the incremental improvement in fit from adding another component appeared to level off. As expected, the reproducibility results showed a largely inverse pattern, with both reproducibility indices decreasing with increasing model orders (Supplementary Fig. 2b). Of note, for model orders up to 8, median spatial correlations across all matched components was above .99, indicating almost identical parcellations derived from the two independent samples. However, even for a model order of 100, the median pairwise correlation was still pretty high (0.85), with 60% of components showing pairwise spatial correlations above 0.8, suggesting a reasonable level of reproducibility even for the most fine-grained parcellation. Our reproducibility index for the categorical parcellations showed very similar results; the adjusted Rand index remained above .9 for model orders up to 8, and then decreased to 0.56 at a model order of 100.

Given that the change in reconstruction error appeared to stabilize between 15 and 30 components, indicating that the intrinsic dimensionality of the data had been approximated, we searched for the most reproducible parcellations within this range. Since the reliability estimates for model orders of 16 and 23 were very similar, with the 23-component solution explaining more of the variance in the original data, we used the 23-component parcellation for all further analyses.

## Adjustment and rank-order normalization of anatomical indices

Prior to being subjected to genome-wide association analyses, we adjusted all anatomical indices for effects of age, sex, estimated total intracranial volume, scanner site, 40 genetic population components, genetic batch and mean Euler number (i.e, an index of MRI image quality[34], averaged across hemispheres), using general additive models. Finally, all adjusted anatomical indices were inverse rank normalized[82].

## Pre-processing of genetic data

For all genetic analyses we made use of the UKB v3 imputed data, which has undergone extensive quality control procedures as described by the UKB genetics team[83]. After converting the BGEN format to PLINK binary format, we additionally carried out standard quality check procedures. We first selected White Europeans, as determined by a combination of self-identification as 'White British' and similar genetic ancestry based on genetic principal components, that had undergone the neuroimaging protocol. We then filtered out individuals with more than 10% missingness, removed SNPs with low imputation quality scores (INFO < 0.5), SNPs with more than 10% missingness, and SNPs failing the Hardy-Weinberg equilibrium test at $p = 1 \times 10^{-9}$. We further set a minor allele frequency threshold of 0.005 leaving 9,061,238 SNPs. After estimating the genetic relationship matrix (GRM) using genetic complex trait analysis (GCTA[36]), we finally removed 910 participants defined as close relatives using a threshold of 0.05 (approximately corresponding to 3rd cousins).

## Heritability estimation and genetic correlation analyses

SNP-based heritability estimates for all morphological features—as well as the pairwise genetic correlations between cerebellar features—were estimated using genetic complex trait analysis (GCTA[36]).

## Univariate genome-wide association analyses

Univariate analyses of total cerebellar grey matter volume and the ten cerebral comparison phenotypes were conducted using Plink v1.9.

## Genetic correlation analyses across discovery and replication samples and with previous published GWASs

Genetic correlation analyses across different samples were conducted using LD-score regression[35].

## Multivariate genome-wide association analyses

For our main analysis, we used a recently developed multivariate analysis method (MOSTest[17]), to conduct a multivariate genome-wide association (GWA) analysis on cerebellar morphological features. MOSTest identifies genetic effects across multiple phenotypes, yielding a multivariate GWAS summary statistic across all 23 features, and provides robust (permutation based) test statistics. For mathematical details of the implementation, see van der Meer et al. (2020)[17], for details on the software implementation see github.com/precimed/mostest. MOSTest has been extensively validated in the original methods paper, including simulations and comparisons with other methods that have confirmed its solid discovery performance as well as an order of magnitude shorter runtime compared to other tools[17]. For comparison to standard univariate approaches, we also performed univariate GWASs (extracted from the univariate stream of MOSTest[17] and identical to results from analyses using Plink).

## Multivariate replication analysis

To ensure that not only single locus lead SNP associations replicate, but that also the multivariate pattern of these associations are consistent in the discovery and replication sample, we implemented a multivariate replication procedure established in Loughnan et al.[41]. In brief, for each locus lead SNP identified in the multivariate analysis in the discovery sample, this procedure derives a composite score from the mass-univariate $z$-statistics and tests for associations of the composite score with the genotype in the replication sample (for mathematical formulation see Loughnan et al.[41]) 12 of the 351 locus lead SNPs could not be tested as they were not available in the replication sample after QC. For the remaining SNPs we report the percent of loci replicating at $P < 0.05$, the percent remaining significant after Bonferroni correction for 339 conducted tests, and the percent of lead SNPs showing the same effect direction.

## Multivariate cerebral comparison phenotypes

To compare key multivariate cerebellar results with other multivariate brain phenotypes, we downloaded summary statistics from two recent studies on cerebrocortical[18] and hippocampal[19] regional morphology. These comparison summary statistics were next analyzed using FUMA[39] as described for the main cerebellar results below.

## Locus identification and SNP annotation

To identify genetic loci we uploaded summary statistics to the FUMA platform v1.4.1[39] Using the 1000GPhase3 EUR as reference panel, we identified independent SNPs as SNPs below the significance threshold of $P < 5e-8$ that were also in linkage equilibrium with each other at $r^2 < 0.6$. For each independent SNP, all candidate variants are identified as variants with LD $r^2 \geq 0.6$ with the independent SNP. A fraction of the independent significant SNPs in approximate linkage equilibrium with each other at $r^2 < 0.1$ were considered as lead SNPs. For a given lead SNP, the borders of the genomic locus are defined as min/max positional coordinates over all corresponding candidate SNPs. Finally, loci are merged if they are separated by less than 250 kb.

FUMA further annotates associated SNPs based on functional categories, Combined Annotation Dependent Depletion (CADD) scores which predicts the deleteriousness of SNPs on protein structure/function[55], RegulomeDB scores which predicts regulatory functions[84]; and chromatin states that shows the transcription/regulation effects of chromatin states at the SNP locus[85]. For all these analyses, we used default FUMA parameters.

## Genome-wide gene-based association and gene-set analyses

We conducted genome-wide gene-based association and gene-set analyses using MAGMA v.1.10[50] (http://ctg.cncr.nl/software/magma). MAGMA performs multiple linear regression to map the input SNPs to 19,190 protein coding genes and estimates the significance value of that gene. Genes were considered significant if the $P$ value was <0.05 after Bonferroni correction for 19,190 genes. The same procedure was used for MAGMA analysis of summary statistics for the three multivariate cerebral comparison phenotypes (cerebrocortical thickness and surface area[18] and regional hippocampal volumes[19]).

MAGMA gene-level statistics were next used as input to gene-property and gene-set analyses in MAGMA. Gene-property analyses test for associations between tissue specific gene expression profiles and disease-gene associations. The gene-property analysis is based on the regression model: $Z \sim \beta_0 + E_t\beta_E + A\beta_A + B\beta_B + \epsilon$, where Z is a gene-based Z-score converted from the gene-based $P$-value, B is a matrix of several technical confounders included by default (e.g., gene size, gene density, sample size), $E_t$ is the gene expression value of a testing tissue type and A is the average expression across all tissue types in a data set (ensuring a test of expression specificity). We performed a one-sided test ($\beta_E > 0$) which is essentially testing the positive relationship between tissue specificity and genetic association of genes.

We tested associations with two regional brain gene expression datasets (Allen Human Brain Atlas[38] and GTEx) and one developmental brain gene expression dataset (BrainSpan). For extraction and processing of gene expression data, see below.

## Extraction and processing of gene expression data

*Allen Human Brain Atlas:* Regional microarray expression data were obtained from 6 post-mortem brains (1 female, ages 24.0–57.0, 42.50 + /− 13.38) provided by the Allen Human Brain Atlas[38]. Data were processed with the abagen toolbox (version 0.1.3)[37] using two volumetric atlases; 1) the binarized 23-region NNMF-derived parcellation of the cerebellar cortex; and 2) a modified version of the Desikan atlas were ROIs were merged to construct 9 bilateral regions: cerebellum, cerebral cortex, pallidum, caudate, putamen, thalamus, amygdala, nucleus accumbens and hippocampus.

First, microarray probes were reannotated using data provided by Arnatkeviciute, Fulcher and Fornito[86]; probes not matched to a valid Entrez ID were discarded. Next, probes were filtered based on their expression intensity relative to background noise[87], such that probes with intensity less than the background in >=50.00% of samples across donors were discarded, yielding 31,569 probes. When multiple probes indexed the expression of the same gene, we selected and used the probe with the most consistent pattern of regional variation across donors (i.e., differential stability[88]), calculated with:

$$\Delta S(p) = \frac{1}{\binom{N}{2}} \sum_{i=1}^{N-1} \sum_{j=i+1}^{N} \rho\left[Bi(p), Bj(p)\right]$$

where $p$ is Spearman's rank correlation of the expression of a single probe, p, across regions in two donors $B_i$ and $B_j$, and N is the total number of donors.

Here, regions correspond to the structural designations provided in the ontology from the AHBA. The MNI coordinates of tissue samples were updated to those generated via non-linear registration using the Advanced Normalization Tools (ANTs; https://github.com/chrisfilo/alleninf). Samples were assigned to brain regions in the provided atlas if their MNI coordinates were within 2 mm of a given parcel. To reduce the potential for misassignment, sample-to-region matching was constrained by hemisphere and gross structural divisions (i.e., cortex, subcortex/brainstem, and cerebellum, such that e.g., a sample in the left cortex could only be assigned to an atlas parcel in the left cortex[86]). All tissue samples not assigned to a brain region in the provided atlas were discarded. \n\nInter-subject variation was addressed by normalizing tissue sample expression values across genes

using a robust sigmoid function[89]:

$$x_{norm} = \frac{1}{1 + \exp\left(-\frac{(x - \langle x \rangle)}{IQR_x}\right)}$$

where $\langle x \rangle$ is the median and $IQR_z$ is the normalized interquartile range of the expression of a single tissue sample across genes. Normalized expression values were then rescaled to the unit interval:

$$x_{scaled} = \frac{x_{norm} - \min(x_{norm})}{\max(x_{norm}) - \min(x_{norm})}$$

Gene expression values were then normalized across tissue samples using an identical procedure. Samples assigned to the same brain region were averaged separately for each donor and then across donors, yielding two regional expression matrices with 23 and 9 rows, corresponding to brain regions, and 15,631 and 15,633 columns, corresponding to the retained genes. Prior to inclusion in MAGMA gene property analyses, we converted gene names for the modified Desikan atlas to ENSMBL IDs and calculated the mean expression value across tissue types (in order to include this as a covariate in MAGMA analyses testing for gene expression specificity), resulting in a 10 (regions) by (15,490) gene expression matrix.

*GTeX:* Text files containing median transcript per millimeter (TPM) values for 53 tissue types (GTEx_Analysis_2017-06-05_v8_RNA-SeQCv1.1.9_gene_median_tpm.gct.gz) were down-loaded from the GTEx portal (https://gtexportal.org/home/datasets/). After selecting only expression data from the seven (out of nine) comparison brain regions present in the GTEx dataset (i.e., amygdala, caudate, cerebellum, cortex, hippocampus, nucleus accumbens and putamen), we filtered the data by only including genes with median TPM values above 1 for at least one of these tissue type (retaining 19,578 of 56,200 annotated genes). Following the procedure used by FUMA[39], we next winsorized median TPM values at 50 (i.e., replaced TPM > 50 with 50), before log transforming TPM with pseudocount 1 (log2(RPKM + 1)). Finally, we calculated the mean expression value across tissue types (in order to include this as a covariate in MAGMA analyses testing for gene expression specificity).

*BrainSpan data:* The analysis of BrainSpan data testing for developmentally specific brain expression was performed entirely within FUMA v1.4.1., using default parameters.

### Analysis of human dating genome data

The Atlas of Variant Age for chromosomes 1-22 was downloaded from the Human Genome Dating (HGD) website: https://human.genome.dating/. This atlas contains more than 45 million SNPs which has been assigned dates of origin based on a recombination clock and mutation clock applied to two large-scale sequencing datasets (1000 Genomes Project[90] and The Simons Genome Diversity Project[91]), with no assumptions made about demographic or selective processes[47]. The current study used the median joint age estimates from both clocks when analyzing SNPs present in both datasets in combination (i.e., 13,694,493 SNPs).

After merging dated SNPs with the 40,405,505 SNPs also present in the Haplotype Reference Consortium reference data (to add minor allele frequencies, MAF) and removing 715,083 (5.2%) SNPs with missing MAF values as well as the very few (14,549, 0.1%) SNPs dated older than 2 million years, 12,960,066 SNPs remained.

548 of these dated SNPs were matched to lead SNPs linked to cerebellar morphology (defined as being in mutual LD at an $r^2$ threshold < 0.1), hereafter referred to as cerebellar-SNPs. Partially because very rare variants (Minor Allele Frequency/MAF < 0.005) had been removed prior to the multivariate GWAS analysis, MAF was not equally distributed between these cerebellar-SNPs and the full range of dated SNPs in the HGD dataset. Importantly, MAF has been shown to be systematically related to the estimated age of SNPs, with a higher proportion of low-MAF SNPs in more recent than in older time-bins[47,48]. Consequently, following the analysis

approach established by Libedinsky et al.[47], we first determined the MAF-distribution of cerebellar-SNPs (across eight bins: <0.0001; 0.0001–0.001; 0.001–0.01; 0.01–0.1; 0.1–0.2; 0.2–0.3; 0.3–0.4; 0.4–0.5) and selected random set of 2,892,270 SNPs from the HGD dataset that were matched to the cerebellar-SNPs in terms of MAF-bin distribution. Similar MAF-matched HDG subsets were created for lead SNPs associated with the two cerebrocortical comparison phenotypes, i.e., regional cerebrocortical surface area and thickness (862 and 714 lead SNPs, respectively).

For statistical inference we constructed null models (separate for each brain phenotype) by randomly drawing sets of SNPs (of equal size to the number of phenotype-linked lead SNPs) from the MAF-matched HGD-datasets and computing the histograms of estimated dates from 0 to 2 million years ago (divided into 100 bins of 20.000 years) over 10,000 iterations. From these null models we extracted bin means as well as significance thresholds (defined as the upper and lower 99.95th percentile of the null model (i.e. corresponding to a two-tailed threshold of 0.05 Bonferroni corrected across 100 bins).

For the validation analyses, we ran the same analyses based on the larger number (range across phenotypes: 1574–2883) of individual significant SNPs (defined as being in mutual LD at an $r^2$ threshold < 0.6).

### Positional and eQTL mapping of SNPs to plausible causal genes

In addition to using MAGMA, we also mapped candidate SNPs to plausible causal genes using two complementary gene mapping strategies implemented ion FUMA[39]: 1) Positional mapping of deleterious SNPs (defined as having a CADD-score > 12.37) and 2) eQTL-mapping of SNPs previously shown to alter gene expression in cerebellar tissue (from the BRAINEAC and GTEx v8 databases). These analyses were run with default FUMA parameters. For the three multivariate cerebral comparison phenotypes (i.e., cerebrocortical thickness, cerebrocortical surface area and hippocampal regional volumes), we employed identical gene mapping procedures to our cerebellar morphology results, except for the tissues chosen for eQTL mapping (cerebrocortical and hippocampal, respectively).

### Gene set analyses using lists of mapped genes

All gene set analyses using mapped genes were conducted using the hypeR R-package[92]. This package implements the hypergeometric test (also known as Fisher's exact test), which assigns a *p*-value to gene-set overlaps given gene set sizes and the number of background genes. This R-package also contains functions for downloading and/or formatting gene sets. The following gene sets were accessed using hypeR: "Allen_Brain_Atlas_up" (regional overexpression in the Allen Muse Brain Atlas), "MGI_Mammalian_Phenotype_Level_4_2021", "DisGeNet" (all from the enrichR platform. In addition, we downloaded sets of genes with regional overexpression in the Allen Human Brain Atlas from the Harmonizome platform[93] and accessed a list of genes mapped to human accelerated regions[94]. For all gene-set analyses we employed Bonferroni-correction by dividing the *p*-value threshold of 0.05 by the number of gene sets included in each analysis.

### Genetic overlap between cerebellar morphology and brain disorders

We accessed GWAS summary statistics for attention deficit hyperactivity disorder (ADHD)[95], autism spectrum disorder (ASD)[96], bipolar disorder (BIP)[97] and major depressive disorder (MDD)[98] from the Psychiatric Genomics Consortium. In order to avoid sample overlap, for MDD we used summary statistics based on a sample with the UK Biobank participants removed. 23&me participants included in the original MDDGWAS were also excluded, since these data are not freely available). Finally, we included data from a recent study of schizophrenia (SCZ)[99]. Shared variants associated with cerebellar morphology and each of the above-mentioned brain disorders were identified using conjunctional FDR statistics (FDR < 0.05)[100,101]. In contrast to genetic correlation analysis, conjunctional FDR does not require effect directions and can therefore be applied to summary statistics from multivariate GWAS, which do not contain effect directions. Two genomic regions, the extended major histocompatibility

complex genes region (hg19 location Chr 6: 25119106–33854733) and chromosome 8p23.1 (hg19 location Chr 8: 7242715–12483982) for all cases and *APOE* region for ASD, were excluded from the FDR-fitting procedures because complex correlations in regions with intricate LD can bias FDR estimation. We further controlled for spurious enrichment by calculating all conditional Q-Q plots after random pruning averaged over 500 iterations. At each iteration, one SNP in every LD block (defined by an $r^2 > 0.1$) was randomly selected and the empirical cumulative distributions were computed using the corresponding *p*-values. Finally, we submitted the results from conjunctional FDR to FUMA v1.3.7[39] to annotate the genomic loci with conjFDR value < 0.10 having an $r^2 \geq 0.6$ with one of the independent significant lead SNPs. Genetic correlations between univariate results for the 23 cerebellar features, the 10 comparison brain phenotypes, and each of the five mental disorders were computed using LD-score regression as described above.

## Data availability

In this study we used brain imaging and genetics data from the UK Biobank [https://www.ukbiobank.ac.uk/], and GWAS summary statistics obtained from the Psychiatric Genomics Consortium [https://www.med.unc.edu/pgc/shared-methods/] and GWAS catalog (https://www.ebi.ac.uk/gwas/). The summary statistics for cerebellar morphology derived in this study are available online in the GWAS Catalog: https://www.ebi.ac.uk/gwas/studies/GCST90728589 to https://www.ebi.ac.uk/gwas/studies/GCST90432154/GCST90728613. FUMA results are available online at https://fuma.ctglab.nl/browse/295626. The Human Genome Dating Dataset (HGD) is available at https://human.genome.dating.

## Code availability

All code and software needed to generate the results is available as part of public resources, specifically FreeSurfer (https://surfer.nmr.mgh.harvard.edu), SUIT, OPNMF (https://github.com/asotiras/brainparts), MOSTest (https://github.com/precimed/mostest), FUMA, MAGMA, abagen (https://github.com/rmarkello/abagen), conjunctional FDR and LD score regression (https://github.com/bulik/ldsc).

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

## Acknowledgements

This work was performed on the Tjeneste for Sensitive Data (TSD) facilities, owned by the University of Oslo, operated and developed by the TSD service group at the University of Oslo, IT-Department (USIT) and on resources provided by UNINETT Sigma2—the National Infrastructure for High Performance Computing and Data Storage in Norway. The research has been conducted using the UK Biobank Resource (access code 27412) and using summary statistics for various brain disorders. We would like to thank the research participants and employees of UK Biobank and the consortia contributing summary statistics for making this work possible. The authors were funded by the South-Eastern Norway Regional Health Authority (TM: 2021-040; OAA: 2013-123, 2017-112, 2019-108. LTW: 2014-097, 2015-073, 2016-083), the Research Council of Norway (TK: 276082, 323961. OAA: 213837, 223273, 248778, 273291, 262656, 229129, 283798, 311993. LTW: 204966, 249795, 273345), Norwegian Health Association (SB: 22731), Stiftelsen Kristian Gerhard Jebsen, and the European Research Council (LTW: ERCStG 802998). The funding bodies had no role in the analysis or interpretation of the data; the preparation, review or approval of the manuscript; nor in the decision to submit the manuscript for publication.

## Author contributions

T.M. originally conceived of the project. T.M., D.v.d.M. and S.B. performed the analyses. T.M., L.T.W. and O.A.A. wrote the initial draft of the manuscript. T.M., Dvd.M., D.R., O.F., S.B., T.K., S.F.-C., M.K., T.W., J.D., O.B.S., A.S., A.D., O.A.A. and L.T.W. discussed the results and contributed to the final manuscript.

## Funding

## Competing interests

The authors declare the following competing interests: O.A.A. has received speaker's honorarium from Lundbeck and is a consultant to HealthLytix. A.M.D. is the founder of and holds equity in CorTechs Labs Inc. and serves on its Scientific Advisory Board. A.M.D. is also a member of the Scientific Advisory Board of Human Longevity Inc. and receives funding through research agreements with General Electric Healthcare and Medtronic Inc. The terms of these arrangements have been reviewed and approved by UCSD in accordance with its conflict-of-interest policies. A.M.D. is an inventor on a patent related to this work, filed by CorTechs Labs Inc. (9 US-7324842B2, filed 22 January 2002, published 29-01-2008). The other authors declare that they have no competing interests.
