## [Transparent Peer Review file · Communications Biology]

The genetic architecture of human cerebellar morphology supports a key role for the cerebellum in recent human evolution and psychopathology

Corresponding Author: Dr Torgeir Moberget

Version 0:

Reviewer comments:

Reviewer #1

(Remarks to the Author)

Thank you for giving me the opportunity to review the manuscript "The genetic architecture of human cerebellar morphology supports a key role for the cerebellum in human evolution and psychopathology" by Moberget and colleagues. In this work, the authors apply multivariate GWAS to cerebellar substructures and identify 351 genetic loci, many novel, and interpret their findings in light of the evolutionary timescale. The paper is well-written and thorough. I do have some questions that center around the question what phenotype the authors are studying, both on a practical and a conceptual level.

- 1) Introduction: paragraph: "Of note.." While I do agree that investigating the cerebellar substructures would potentially provide more insight compared with investigating the cerebellum as a whole, the approach the authors take here (and present as their main result) is essentially the genetics of what is common amongst all those regions. In that light, this paragraph is a bit odd. Based on this reasoning, wouldn't it be interesting to investigate the signals that are NOT shared?
- 2) This question is more general, but since many authors are shared between this paper and the MOSTest paper: could the authors provide an intuitive reason why MOSTest provides up to 35x as many results (as reported here?). Given that the individuals included in this analysis are the same for all phenotypes, my naive idea would be that the increase in power should stem from a better separation of signal and noise, and maybe by combining all the substructures, the errors will cancel out. But given that smaller structures usually have lower test-retest reliability, so probably new noise it introduced by the segmentation, I still find it hard to understand where the increase in power comes from. Do the authors have an intuitive reason for this?
- 3) The main paper (results, line 102 and onwards) does not discuss what it is that the GWAS is performed on: the methods discussed the SUIT toolbox to create 28 cerebellar (sub)volumes, but if I understand correctly, these values are only used for QC purposes. Maybe I missed it, but the methods speak of "anatomical indices" (line 785) which are not defined: is it the average/sum of values in the VBM maps per component? If not, what were the 23 phenotypes in the multivariate analysis?
- 4) Based on the three questions above, I would like a discussion on what trait is investigated in this multivariate analysis and how the reader should interpret the genetic signal that is found.

Other points:

- 5) Figure 2: without numbers panels B-D are really hard to interpret. Same for Fig 7A-B.
- 6) Results, line 220: "Thus, 329 (94%) of the 351 reported loci 219 were replicated." To me this number does not follow from what is stated above. How do the authors define replication? I think this statement is based on methods, line 841: "In addition, we report the percentage of genetic loci replicated (defined as containing at least one replicating lead SNP)." Please clarify in the main text (it is a key finding, mentioned in the abstract, so should be clear).
- 7) Results, line 233: "52 of the 57 loci identified in the univariate analyses of regional cerebellar features overlapped 55 of

the 351”; this is unclear to me: how can 52 loci overlap with 55?

8) Results, line 266: “and of the mean Z-score across these features (i.e. analogous to the main effect on overall cerebellar volume)”. I am not sure I agree with this interpretation: the mean Z-score could be driven by the extremes (i.e. effects in one of the subvolumes only).

9) Methods (line ~684): Why remove data at $> 3 * |MAD|$? Assuming that the brain measures are approximately normal (which I assume they are); if I am not mistaken this would be equivalent at removing outliers at $|Z| > 2$, which is about 5% of the data and with these large sample sizes amounts to at least half of the participants that were removed for the global measures. Indeed, the rejection of 15.3% of datasets seems rather strict?

10) Methods (line 788): “and a quantitative structural MRI quality index (the Euler number);” this would suggest there is a linear association between quality and e.g. volume? Is that reasonable to assume?

11) SI table 3; please update GWAS sources

12) Methods: line 984: “Are the hypergeometrical tests taking gene-correlations into account?”

13) While I appreciate the comparison with other MOSTest phenotypes in the results, it is not fully clear why this was done; From the discussion: “While the comparisons with cerebral features should be interpreted with care since different image preprocessing pipelines were employed,...”. Can the authors explain this remark? Assuming that image processing/segmentation was done reliably, why would the choice of pipeline influence the genetic results? If the authors expect an influence of pipeline, why then this comparison?

Reviewer #2

(Remarks to the Author)

Moberget and colleagues describe multivariate GWAS analyses on cerebellar volumetric measures in the UK Biobank (total 38K individuals with MRI scans). First, they applied a data driven parcellation procedure of the cerebellum. Next they ran GWAS on the 23 regions and combined these in a multivariate GWAS framework (MOSTest). They identified 351 loci with significant effects in the multivariate analysis. These loci are enriched for genes expressed in the cerebellum, those involved in neuronal projection/development, and psychiatric traits.

The authors have performed a wide range of analyses and present these in a clear and comprehensive way. Overall, I think this is a good paper and most analyses are in line with current knowledge/hypotheses on the cerebellum. I have some remarks and aspects that could be improved.

The multivariate framework increases power to find associated SNPs, but comes at the cost of losing information and spatial resolution. As a result, MV GWAS results are harder to interpret and less suitable for most downstream analyses to gain insight in biology of cerebellar development and disease.

First, the MV approach lumps all association signals together into a single p-value. In contrast to the cerebral hemispheres, much less is known about the spatial/functional organization of the cerebellum, but this information is lost in the MV framework. Second there is no overall direction of effect (beta coefficient) that can be used for other analyses such as LD-score regression or Mendelian Randomization. The authors adopt other enrichment methods, which can have some caveats (see below).

The authors state there is no inflation in the MOSTest analyses. But they give no lambdaGC value, the QQ plot is scaled in a way that inflation cannot be assessed. Can the authors provide a LambdaGC?

Furthermore, it is not totally clear how the authors obtained the univariate cerebellar GWAS results. The phenotypes are residualized on the covariates in MOSTest. Did the authors run “regular” univariate GWAS (PLINK?) on these residualized outcomes? Can they show there is no residual confounding? QQ-plots and LDSC intercept for these univariate GWAS could be provided to ensure there is no/limited residual confounding.

The univariate GWAS on total cerebellar grey matter and cerebral volumes were performed in PLINK. Which model was used, which covariates? Again, what was the genomic inflation factor and LDSC intercept?

In the GCTA h2 estimates, how many PCs were included?

The authors use permutations to show enrichment for “recent” genetic variation (fig 4). The randomly selected SNPs are matched for allele frequency with brain related SNPs, but sampling 41,549 random SNPs confined within the MAF bins might be too random. This disregards the correlation structure (LD) between the 41,549 SNPs. SNPs in high LD are likely to have the same “age”. Will the results still hold when the authors select independent SNPs from the GWAS: only top hits or at least stringently LD-pruned SNPs only?

It could be interesting to run single cell enrichment analyses using FUMA as well: <https://fuma.ctglab.nl/celltype>.

The authors use conditional/conjunctional FDR to assess and leverage a genetic overlap between psychiatric diseases and the cerebellum. Can't this be biased due to SNP LD-score? Those SNPs with high LD scores are more likely to have higher

Chisq statistics for both traits if they tag more genetic variation and thus conditional QQ-plots show the inflation pattern.

Will the authors release the univariate and multivariate summary statistics for the public?

Minor points:

Fig 3: there is no colour legend for the colours in the cerebellum (Z-scores probably?).

Fig 5C: developmental stages could be ordered from first to last instead of $-\log_{10}(P)$ to make it more intuitive.

Version 1:

Reviewer comments:

Reviewer #1

(Remarks to the Author)

The revised version of the manuscript "The genetic architecture of human cerebellar morphology supports a key role for the cerebellum in human evolution and psychopathology" by Moberget and colleagues reads well and answered most of the questions I had while reading the first version. I only have a few minor points left.

1) In answer to my previous point 1 the authors do explain why they chose to investigate the cerebellum through multivariate analysis (essentially, to pick up moderate bidirectional effects), the introductory paragraph ("Of note.." still discusses the separate parts of the cerebellum with respect to their separate functions). Please add a sentence that justifies the multivariate approach to help the reader.

2) I do appreciate the addition of the univariate results for the cerebellar covariance patterns, but I do still wonder about the addition of the "comparison phenotypes". The authors explain their rationale better than the in the previous version of this manuscript, but I feel it could be even further simplified.

"To validate our analysis approach, we computed genetic correlations (using LD-score regression, LDSC32) between univariate GWAS results on the comparison features (see Manhattan and QQ-plots in Supplementary Figure 5) and previously published neuroimaging GWAS studies on these brain phenotypes.

If I am totally honest, I still do not see the purpose of this analysis. The fact that these overlap could stem from the fact that many of the previously published data also includes (part of) the UKB data and this does not validate the approach for the main paper. Why not leave out all the phenotypes that are not truly used (as far as I can see, the phenotypes that are explicitly compared to the multivariate analysis are hippocampus, Surface Area and Cortical thickness)?

3) Around line 205: Maybe add a sentence on that/how the univariate GWAS were run; the current description jumps from preprocessing the phenotypes to SNP heritability.

4) Line 236: Supplementary Data 8 should probably read Supplementary Data 6.

5) Supplementary Data: The table descriptions contain some typos e.g. thickness (SI6) mapping strategies (SI22-25).

Reviewer #2

(Remarks to the Author)

I thank the authors for their comprehensive reply and thorough additional analyses. I have no further remaining questions or remarks.

Reviewers' comments:

Reviewer #1 (Remarks to the Author):

Thank you for giving me the opportunity to review the manuscript “The genetic architecture of human cerebellar morphology supports a key role for the cerebellum in human evolution and psychopathology” by Moberget and colleagues. In this work, the authors apply multivariate GWAS to cerebellar substructures and identify 351 genetic loci, many novel, and interpret their findings in light of the evolutionary timescale. The paper is well-written and thorough. I do have some questions that center around the question what phenotype the authors are studying, both on a practical and a conceptual level.

Response: *We thank the reviewer for this positive evaluation of our work and for the constructive and thorough suggestions for further improvements and clarifications.*

1) Introduction: paragraph: “Of note..” While I do agree that investigating the cerebellar substructures would potentially provide more insight compared with investigating the cerebellum as a whole, the approach the authors take here (and present as their main result) is essentially the genetics of what is common amongst all those regions. In that light, this paragraph is a bit odd. Based on this reasoning, wouldn't it be interesting to investigate the signals that are NOT shared?

Response: *We agree with the reviewer that our motivation for choosing a multi-variate analysis of cerebellar substructures should be further explained. From a conceptual point of view, we chose a multivariate approach since genetic variants have previously been shown to have distributed effects across brain regions and morphological brain features (see for instance van der Meer et al, Nature Communications, 2020¹). Importantly, such distributed genetic effects are not necessarily unidirectional (e.g., associated with increased volume across the entire cerebral cortex), but might also show regionally specific effects of opposite polarity. Such bidirectional genetic effects could potentially sum to zero in an averaged global measure, and thus be invisible to GWASs examining global measures, such as total cerebrocortical volume. Indeed, many such patterns were described in the original MOSTest paper (see also Figure 3 in the current manuscript for some examples of such patterns in the cerebellum). While some of the strongest regionally specific effects would indeed emerge as significant also in univariate analyses of regional morphological features, this approach would not be very sensitive to moderately strong distributed effects (due to its inability to integrate effects across features). Since our overall aim in this study was to map genetic variants influencing the main features of inter-individual variation in cerebellar morphology (i.e., including regional as well as global features), we chose the multivariate MOSTest approach for our analyses. While we agree that a specific focus on regional cerebellar features (e.g., while adjusting for total cerebellar volume) might indeed also be interesting, we believe this question falls beyond the main scope of the current paper.*

We nonetheless agree with the Reviewer that the univariate analyses are also of interest. Thus, we will also make the univariate summary statistics publicly available to the research community and have added a set of Supplementary Figures (4-6) showing Manhattan- and QQ-plots for all univariate GWAS analyses (together with the LDSC-intercept as a test of genetic inflation, see below). Of note, as seen in Figure 2B, most lead SNPs from the multivariate GWASs were not significant in the univariate analyses, demonstrating the boost in power provided by our multivariate approach.

Supplementary Figure 4: Manhattan and QQ-plots for univariate GWASs on cerebral comparison phenotypes in the current sample. Each plot also provides the intercept from LD-score regression analyses (LDSC-intercept), where values close to 1 indicate no or minimal inflation. Manhattan- and QQ-plots from FUMA (<https://fuma.ctglab.nl/>).

Supplementary Figure 5: Manhattan and QQ-plots (with LDSC-intercepts) for univariate GWASs on cerebellar structural covariance patterns (SCPs) 1-14. Manhattan- and QQ-plots from FUMA (<https://fuma.ctglab.nl/>).

Supplementary Figure 6: Manhattan and QQ-plots (with LDSC-intercepts) for univariate GWASs on cerebellar structural covariance patterns (SCPs) 15-23. Manhattan- and QQ-plots from FUMA (<https://fuma.ctglab.nl/>).

Moreover, individual feature Z-scores for all 351 discovery sample loci lead SNPs from the multivariate analysis can be found in Supplementary Table 10. These Z-scores show the distribution of effects across the individual cerebellum, thus recovering some of the spatial information that is lost when integrating across all features.

Finally, to better clarify our motivation for a multivariate approach (as well as the specific features of the multivariate analysis method), we have now made several changes to the manuscript (see below, after our response to comment 4).

2) This question is more general, but since many authors are shared between this paper and the MOSTest paper: could the authors provide an intuitive reason why MOSTest provides up to 35x as many results (as reported here?). Given that the individuals included in this analysis are the same for all phenotypes, my naive idea would be that the increase in power should stem from a better separation of signal and noise, and maybe by combining all the substructures, the errors will cancel out. But given that smaller structures usually have lower test-retest reliability, so probably

new noise it introduced by the segmentation, I still find it hard to understand where the increase in power comes from. Do the authors have an intuitive reason for this?

Response: *In short, MOSTest on regional features identifies up to 35 times as many loci as traditional GWAS on total cerebellar volume primarily for two reasons. First, MOSTest can pick up the effects of variants influencing just a few select regional features (e.g., rs78777685 in Figure 3), effects that may largely disappear when considering the cerebellum as a uniform anatomical feature. Indeed, many variants with large effects on specific cerebellar features (including rs78777685) were also discovered in the univariate regional analyses, which also yielded increased discovery relative to the univariate analysis of total cerebellar volume. Second, many genetic variants were associated with patterns of opposite – and relatively modest – effect directions (consistent across discovery and replication samples) across regional features (e.g., rs2388334 in Figure 3). These effects are not strong enough for any single feature to be identified in the univariate analyses, and effectively summing to values close to zero when averaged across the entire cerebellum (so not noticeable in the univariate analysis of total cerebellar volume). We also refer to Figure 3 in the manuscript for some additional example patterns. The response to comment 1 above also provide some explanation of the reason for boost in power.*

3) The main paper (results, line 102 and onwards) does not discuss what it is that the GWAS is performed on: the methods discussed the SUIT toolbox to create 28 cerebellar (sub)volumes, but if I understand correctly, these values are only used for QC purposes. Maybe I missed it, but the methods speak of “anatomical indices” (line 785) which are not defined: is it the average/sum of values in the VBM maps per component? If not, what were the 23 phenotypes in the multivariate analysis?

Response: *We agree that this was not made sufficiently clear in the Results and Discussion sections and we have now made several changes to improve the clarity of the manuscript. Briefly, we performed the multivariate GWAS on 23 morphological features that were empirically derived from the voxel-based morphometry (VBM) based maps of cerebellar grey matter volume using Non-Negative Matrix factorization (NNMF). Our motivation for choosing this data-driven approach to parcellate voxel-wise cerebellar grey matter maps was primarily based on the observation that traditional anatomical parcellations of the cerebellum (i.e., cerebellar lobules, based on the main fissures) do not correspond well with functionally defined cerebellar regions (see King et al, 2019²). Further, we chose NNMF because this algorithm has been shown to yield sparse and easily interpretable parcellations of brain imaging data³⁻⁷. In essence, when applied to voxel-wise indices of grey matter volume, NNMF yields distinct maps of voxels that show similar patterns of volume-variation across individuals. Such maps are commonly referred to as “structural covariance patterns”^{3,5,6}, and we have used this term in the revised version of the manuscript. For each structural covariance pattern (common across all participants), NNMF also provides individual subject weights expressing the degree to which these patterns are expressed. When applied to voxel-wise cerebellar volumetric maps, as in the current project, these subject weights essentially express individual volumes of the cerebellar sub-region covered by*

the structural covariance maps. Thus, NNMF provides a parts-based segmentation of the cerebellum, based on the observed covariance across voxels. Importantly, when subject weights were summed across the 23 structural covariance maps, the resulting values correlated tightly ($r = 0.9995$) with estimates of total cerebellar grey matter volume (see also Supplementary Figure 4), demonstrating that our data-driven decomposition preserved inter-individual variation in cerebellar volume.

Supplementary Figure 4: The sum of cerebellar NNMF subject weights correlates tightly with total cerebellar grey matter volume. This demonstrates that the subjects weight resulting from the data-driven decomposition using NNMF (reflecting the degree to which a particular cerebellar structural covariance pattern is expressed) preserves inter-individual variation in cerebellar volume.

To better clarify both our motivation for choosing this analysis approach and general features of the chosen algorithm, we have now added this to the Results section (see below, after our response to comment 4).

4) Based on the three questions above, I would like a discussion on what trait is investigated in this multivariate analysis and how the reader should interpret the genetic signal that is found.

Response: We agree with the reviewer that a more in-depth description and discussion of the analyzed anatomical features and the multivariate analysis method used is in order. Consequently, we have now made the following addition to the Results section of the manuscript (pages 3-4):

“Since traditional atlases of the cerebellar cortex based on gross anatomical landmarks (i.e., lobules) only partially overlap with more recent functional parcellations of the cerebellum⁸⁻¹⁶, we first used a data-driven approach (non-negative matrix factorization, NNMF⁵⁻⁷) to parcellate **voxel-based morphometry (VBM) based maps of cerebellar grey matter volume (containing 147,121 1*1*1 mm voxels) from 28,212 participants into robustly reproducible**

sub-regions. Similarly to other dimensionality reduction methods, such as principal component analysis (PCA) and independent component analysis (ICA), NNMF decomposes the input matrix (here 147k voxels * 28k participants) into two lower rank matrices (voxel weights: W , and participant weights: H), so that the product of W and H approximates the original input data. The defining feature of NNMF is that it requires these lower rank matrices to be non-negative, which has previously been shown to result in sparse, and easily interpretable, parcellations of high-dimensional brain imaging data³⁻⁷). In essence, when applied to voxel-wise indices of cerebellar grey matter volume, NNMF yields distinct maps of voxels that show similar patterns of volume-variation across individuals, hereafter referred to as cerebellar “structural covariance patterns”^{3,5,6}.” For each structural covariance pattern (common across all participants), NNMF also provides individual subject weights expressing the degree to which these patterns are expressed.

In the current study NNMF yielded highly reproducible cerebellar structural covariance patterns (across split-half datasets) for model orders (i.e., number of components/patterns specified) ranging from 2 to 100 (see Supplementary Figures 2 for summary maps of all tested model orders). After observing that the improved fit to the original data seen with higher model orders tended to level off between 15 and 30 components (indicating that the intrinsic dimensionality of the data might have been reached (see Supplementary Figure 3), we decided on a model order of 23 based on its good split-half reproducibility (see Fig. 1B, and Online Online Methods for details regarding model order selection). Importantly, when subject weights were summed across the 23 structural covariance maps, the resulting values correlated tightly ($r = 0.9995$) with estimates of total cerebellar grey matter volume (see also Supplementary Figure 4), demonstrating that our data-driven decomposition preserved inter-individual variation in cerebellar volume.”

Other points:

5) Figure 2: without numbers panels B-D are really hard to interpret. Same for Fig 7A-B.

Response: We agree. Key numbers have now been added to all Euler diagrams (see revised Figure 2 below). While we have refrained from numbering all areas of overlap, as this would clutter the figure without adding all that much information, all this information for the cerebellar candidate SNPs can be found in Supplementary Table 8. We have now alerted the reader to this in the legend of Figure 2.

Figure 2: Multivariate GWAS analysis of the 23 cerebellar morphological features revealed 351 independent genome-wide significant (GWS) loci. **A:** The upper half of the Miami plot shows the main results from the multivariate analysis. The lower half displays results from a series of 23 univariate analyses (corrected for multiple comparisons using the standard min-P approach), as well as results from a univariate analysis of total cerebellar grey matter (marked in orange). **B-D:** Euler diagrams showing the relative numbers of - and overlaps between - candidate SNPs (in thousands) mapped by the three analysis approaches employed in the current study (**B**), the current and four recent studies reporting genetic associations with cerebellar morphology (**C**), as well as results from multivariate GWASs on hippocampal and cerebrocortical morphology (**D**). For full results on all cerebellar candidate SNPs, see Supplementary Data 8.

6) Results, line 220: “Thus, 329 (94%) of the 351 reported loci were replicated.” To me this number does not follow from what is stated above. How do the authors define replication? I think this statement is based on methods, line 841: “In addition, we report the percentage of genetic loci replicated (defined as containing at least one replicating lead SNP).” Please clarify in the main text (it is a key finding, mentioned in the abstract, so should be clear).

Response: We thank the reviewer for highlighting our description of the chosen replication strategy, which needs some clarification and correction. In short, we assessed replicability of the 351 locus lead SNPs from the discovery sample (minus 12 SNPs which were not present in the replication sample after QC). For each of these 339 replication sample SNPs, we derived a multivariate composite score, reflecting the degree to which alleles expressed

the same multivariate pattern as in the discovery sample, and tested for effects of this composite score across alleles.

We have now revised the relevant part of the Results section (page 8, lines 238-248), in order to clarify this. The updated version reads as follows (changes highlighted in red):

«We evaluated the robustness of these multivariate results using a multivariate replication procedure established in Loughnan et al¹⁷, which computes a composite score from the mass-univariate z-statistics (i.e., applying multivariate weights from the discovery sample to the replication sample input data) and then tests for associations between this composite score and genotypes in the replication sample (for mathematical formulation see Loughnan et al¹⁷). Results showed that 97% of the 339 loci lead SNPs present in both samples (i.e., 94% of the 351 reported loci lead SNPs from the discovery sample) replicated at a nominal significance threshold of $p < .05$ (Supplementary Figure 9 and Supplementary Data 10), and that 74% remained significant after Bonferroni correction (Supplementary Data 10). Moreover, 99% of loci lead SNPs showed the same effect direction across discovery and replication samples (Supplementary Figure 9 and Supplementary Data 10).»

In addition, we removed the sentence in the Materials and Methods section stating the following: “In addition, we report the percentage of genetic loci replicated (defined as containing at least one replicating lead SNP)”, as this reflects a previous version of the replication analysis that included all 560 lead SNPs (instead of only the 351 locus lead SNPs used in the current version).

7) Results, line 233: “52 of the 57 loci identified in the univariate analyses of regional cerebellar features overlapped 55 of the 351”; this is unclear to me: how can 52 loci overlap with 55?

Response: *We agree that this may seem unclear and have thoroughly examined the loci discovered using the three methods and their overlaps. This examination discovered an error in the original manuscript, as the number of loci from the univariate analyses overlapping multivariate loci was 56, not 52 as originally reported (all other numbers were correct). The explanation for the remaining slight mismatch (56 univariate loci overlapping 55 multivariate loci) is that loci identified by different strategies are not necessarily of equal length, and that one larger locus discovered by one method might thus overlap several smaller loci discovered by another method. For this reason, we used candidate SNPs for the main overlap analyses reported in Figures 2 B-D, but also report number of loci (and their overlap), since this is a conventional way of summarizing GWAS results.*

In addition to correcting the error, we have now made a small addition to the relevant paragraph of the Results section to clarify the reason for this apparent conundrum (addition highlighted in red):

«56 of the 57 loci identified in the univariate analyses of regional cerebellar features overlapped 55 of the 351 loci identified using the multivariate method, while 9 out of 10 loci identified in the univariate analysis of total cerebellar volume were also identified in the multivariate analysis. *Note that the investigated genomic regions are the same, despite slightly mismatching numbers of loci between the approaches. This discrepancy is due to the larger number of genome-wide-significant SNPs in the multivariate analysis, causing some larger loci in the multivariate analyses to overlap with several smaller loci in the univariate analyses.*»

8) Results, line 266: “and of the mean Z-score across these features (i.e. analogous to the main effect on overall cerebellar volume)”. I am not sure I agree with this interpretation: the mean Z-score could be driven by the extremes (i.e. effects in one of the subvolumes only).

Response: Yes, this could indeed in theory be the case, especially for the smaller regional features, where even very large local effects (yielding high Z-scores that clearly influence the mean z-score) might still only minimally influence total cerebellar volumes. Ultimately, then, the degree to which the mean z-score is analogous to total cerebellar volume is an empirical question. Judging from the data presented in Figure 3, our analogy does seem to hold, since all the loci lead SNPs with the most extreme (positive or negative) mean Z-scores were also identified in the univariate analyses of cerebellar volume. In addition, we computed the correlation (across 27,302 individuals) between the mean of the 23 component subject weights and total cerebellar volume (after adjustment for covariates) and plotted the scatter plot of this association. As can be seen below, these measures were highly linearly correlated (Pearson $r = 0.999$):

Thus, we believe our analogy between the mean regional z-score and total cerebellar volume is justified by empirical observations of the data.

9) Methods (line ~684): Why remove data at $> 3*|MAD|$? Assuming that the brain measures are approximately normal (which I assume they are); if I am not mistaken this would be equivalent at removing outliers at $|Z| > 2$, which is about 5% of the data

and with these large sample sizes amounts to at least half of the participants that were removed for the global measures. Indeed, the rejection of 15.3% of datasets seems rather strict?

Response: *Given that manual QC of such a large dataset was unfeasible, we opted for a relatively strict exclusion criterion for automatic outlier rejection. We also note that the same QC-threshold based on 3 MAD have previously been used when analyzing very large neuroimaging datasets (e.g., Tisink et al., Communications Biology 2022¹⁸; Alnæs et al, PNAS 2020¹⁹).*

An additional reason for preferring to err on the side of excessive caution rather than being too permissive of potentially noisy data was our strategy of using a data-driven decomposition algorithm (NNMF) to derive our regional features. Since the principle of “garbage-in-garbage-out” applies to all data-driven de-composition methods, we wanted to make sure that the MRI-based features (i.e., cerebellar structural covariance patterns) to be analyzed further were derived from good quality data.

10) Methods (line 788): “and a quantitative structural MRI quality index (the Euler number);” this would suggest there is a linear association between quality and e.g. volume? Is that reasonable to assume?

Response: *Our use of the Euler number as a linear co-variate partially controlling/adjusting for variation in MRI data quality was based both on published recommendations to include covariates reflecting variation in data quality (e.g., Rosen et al. 2018²⁰, Alfaro-Almogro et al., 2021²¹) and previous experience from our own large-scale imaging studies. We typically observe that the mean Euler number is indeed significantly associated with many neuroimaging-based phenotypes. As an example, the plots below show associations between the mean Euler number and two brain phenotypes in the current discovery sample of 27,302 UK Biobank participants. As can be seen, the mean Euler number shows weak positive correlations with both phenotypes, indicating that estimates of mean cerebrocortical thickness and total cerebellar volume increase slightly with this proxy for increasing MR image quality. We thus consider it an important co-variate to add to the analyses (together with scanner site, genetic batch, sex, age, etc.).*

11) SI table 3; please update GWAS sources

Response: *We thank the reviewer for noticing this unfortunate omission and have updated all GWAS references in SI table 3 the revised manuscript.*

12) Methods: line 984: “Are the hypergeometrical tests taking gene-correlations into account?”

Response: *No, the hypergeometric test (also known as Fisher’s exact test) simply assigns probability values to 2*2 contingency tables listing the genes associated and not associated with a phenotype versus the genes contained in or not contained in a gene set, under the null hypothesis of no enrichment; (i.e., assuming that these two categorical variables are independent). That is, significant results reflect conditions where we observe more genes associated with both the brain phenotype and the gene set tested than would be expected by chance.*

Hypergeometric tests are thus the simplest form of enrichment tests available, and have been criticized for both ignoring differences in the strength of association (they require a hard cut-off of inclusion or exclusion in gene lists) as well as for making the unrealistic assumption of independence between genes²². The first of these criticisms is addressed by MAGMA²³, which the full range of p-values assigned to all protein coding genes overlapping available SNPs, while the second – closely related to the reviewer’s question – is not. We agree that the assumption of gene independence (both with respect to association to brain phenotypes and to curated gene sets) is highly unrealistic, and might indeed make these tests less conservative than commonly assumed²².

However, the simplicity of these tests also come with some notable benefits. First, they allow for testing lists of genes associated to a brain phenotype of interest using different methods (e.g., MAGMA, positional mapping or eQTL mapping), where integration of association strength across methods is impossible. Second, they allow for testing a vast number of available gene sets (like MAGMA) and provide a widely used test statistic that is implemented in standardized platforms for post-GWAS analyses, such as FUMA²⁴.

We further believe that the pattern of results from the hypergeometric tests – especially when converging with results from the MAGMA gene property and gene-set analyses – provide both 1) an important biological validation of the current multivariate GWAS results (by showing the most significant overlap with genes preferentially expressed in cerebellar tissue and linked to cerebellar abnormalities in mouse gene perturbation experiments), and 2) suggest cerebellar involvement in a wide range of human disorders and phenotypes (including schizophrenia and intelligence)

The known limitations of hypergeometric tests notwithstanding, we have thus decided on keeping these analyses and results in the revised version of the manuscript.

13) While I appreciate the comparison with other MOSTest phenotypes in the results, it is not fully clear why this was done; From the discussion: “While the

comparisons with cerebral features should be interpreted with care since different image preprocessing pipelines were employed,...". Can the authors explain this remark? Assuming that image processing/segmentation was done reliably, why would the choice of pipeline influence the genetic results? If the authors expect an influence of pipeline, why then this comparison?

Response: *We agree that this point needs clarification. Our main motivation for including these comparison analyses was to examine the specificity of any cerebellar findings relative to other comparison phenotypes (i.e., whether effects are cerebellum-specific or rather apply generally across the brain). We still believe that this is an important and valuable part of the current paper and have thus decided to keep these comparison analyses. However, we agree that the sentence highlighted in the Discussion is a bit misleading. Our intention was to inform the reader about potential differences between the studies that may influence the comparison. However, since processing pipelines were both tailored to the brain phenotype in question (i.e., the cerebral cortex and the hippocampus). and have been widely used in the literature, we have in the revised version of the manuscript rephrased this sentence, removing the caveat and highlighting the brain phenotype comparison.*

The full sentence now reads (*change highlighted in red*):

"When comparing this finding with results from other multivariate brain phenotypes, we observed that HAR gene enrichment was nominally stronger for cerebellar morphology than for vertex-wise cerebrocortical thickness and area, and significantly stronger than for hippocampal regional volumes (Figure 5D)."

Reviewer #2 (Remarks to the Author):

Moberget and colleagues describe multivariate GWAS analyses on cerebellar volumetric measures in the UK Biobank (total 38K individuals with MRI scans). First, they applied a data driven parcellation procedure of the cerebellum. Next they ran GWAS on the 23 regions and combined these in a multivariate GWAS framework (MOSTest). They identified 351 loci with significant effects in the multivariate analysis. These loci are enriched for genes expressed in the cerebellum, those involved in neuronal projection/development, and psychiatric traits.

The authors have performed a wide range of analyses and present these in a clear and comprehensive way. Overall, I think this is a good paper and most analyses are in line with current knowledge/hypotheses on the cerebellum. I have some remarks and aspects that could be improved.

Response: *We thank the reviewer for this overall positive evaluation of our work and for the constructive and thorough suggestions for further improvements and clarifications.*

1. The multivariate framework increases power to find associated SNPs, but comes at the cost of losing information and spatial resolution. As a result, MV GWAS results are harder to interpret and less suitable for most downstream analyses to gain insight in biology of cerebellar development and disease. First, the MV approach lumps all association signals together into a single p-value. In contrast to the cerebral hemispheres, much less is known about the spatial/functional organization of the cerebellum, but this information is lost in the MV framework. Second there is no overall direction of effect (beta coefficient) that can be used for other analyses such as LD-score regression or Mendelian Randomization. The authors adopt other enrichment methods, which can have some caveats (see below).

Response: *We agree with Reviewer that some spatial resolution is indeed lost when conducting multivariate, relative to univariate, analyses. A similar point was also made by Reviewer #1 (point 1), While we believe a full set of analyses focusing on the univariate results falls beyond the scope of the current paper, we will make all univariate results publicly available upon publication of the manuscript. Moreover, we have now added a new set of Supplementary Figures showing Manhattan plots, QQ-plots and LDSC-intercepts for all these univariate results (Supplementary Figures 4-6), which should aid the interested scientist in performing an initial assessment of the univariate summary statistics. These figures are included in our response to Reviewer #1 (point 1) above.*

We further agree with Reviewer that the lack of directional effects is an important limitation of the multivariate approach and for this reason included some analyses on univariate results (LDSC-based genetic correlations) to further examine the analyses showing genetic overlap between the multivariate signal and genetic risk for developmental/psychiatric disorders in the original manuscript.

We would also like to note, however, that the substantially increased statistical power yielded by the multivariate approach also comes with some important strengths with respect to gaining insights into the biology of cerebellar development and disease. For instance, while the MAGMA- and gene-set analyses based in the multivariate summary statistics yielded several significant findings highlighting relevant biological processes and pathways, similar analyses using univariate results (both from the current and previous studies) tended to yield much weaker – and fewer – associations.

*Thus, we have decided to stick with the multivariate approach for the current paper, but have revised the limitations section (page 22, lines 757-774), to alert the reader to these limitations (**additions marked in red**):*

*“The main limitations of the current study concern the ancestral homogeneity of the sample, the sample size, the exclusion of very rare genetic variants **and the limitations on follow-up analyses placed by multivariate (relative to univariate) test-statistics**. Limiting the sample to participants of European ancestry was deemed necessary considering the current state of the multivariate GWAS methods used but may limit the*

generalizability of our findings. Second, while the current sample size is large in comparison with previous imaging genetics studies, it is still relatively small in comparison to GWASs of other complex human phenotypes (e.g., intelligence, with a current n of > 3 million²⁵). Moreover, very rare genetic variants ($MAF < 0.005$) were excluded from the current multivariate GWAS, but are likely to include a number of variants with relatively large effect sizes on complex human traits²⁶. Thus, future studies using larger and more diverse samples – as well as whole exome and/or genome sequencing – are likely to improve discoverability and refine the characterization of the polygenic architecture associated with cerebellar structure. **Finally, while multivariate GWAS increases the power to detect genetic variants associated with brain phenotypes relative to univariate approaches, it also places limitations on the possible follow-up analyses (e.g., genetic correlations or Mendelian randomization), which require the directional effects. Although comprehensive follow-up analyses of univariate GWAS results fall beyond the main scope of the current report, we have included all summary statistics (univariate and multivariate) in the supplement.**”

2. The authors state there is no inflation in the MOSTest analyses. But they give no lambdaGC value, the QQ plot is scaled in a way that inflation cannot be assessed. Can the authors provide a LambdaGC?

Response: We thank the reviewer for noticing this omission - our claim that there is no inflation in the MOSTest analyses clearly needs to rest on more solid evidence. LambdaGC values have been shown to be unable to distinguish inflation due to bias (such as cryptic relatedness and population stratification) from inflation due to true polygenicity, especially in well-powered GWASs (see e.g., Bulk-Sullivan et al., Nature Genetics 2015²⁷). Following the suggestion by Bulk-Sullivan et al. 2015²⁷, we have used the LDSC intercept as a test of genetic inflation, since this provides a more accurate and robust quantification of the extent of the confounding bias from inflation than LambdaGC.

For the MOSTest summary statistics, the intercept was 1.0352, which suggests minimal to no genetic inflation due to bias (such as cryptic relatedness and population stratification).

This number has now been added to the results section (page 6, line 201) and replaces the QQ-plot as our test for genetic inflation.

We have also provided the LDSC intercept values for all other univariate GWASs performed in the current paper in Supplementary Figures 4-6, as well as in the Supplementary Table 5. These ranged from 0.9963 to 1.0288, indicating no or minimal genetic inflation due to bias.

3. Furthermore, it is not totally clear how the authors obtained the univariate cerebellar GWAS results. The phenotypes are residualized on the covariates in MOSTest. Did the authors run “regular” univariate GWA(S (PLINK?)) on these residualized outcomes? Can they show there is no residual confounding? QQ-

plots and LDSC intercept for these univariate GWAS could be provided to ensure there is no/limited residual confounding.

Response: All genetic analyses, including univariate GWASs, were run on morphological features that were pre-residualized for effects of scanner site, sex, age, estimated total intracranial volume, 40 genetic population components, genetic analysis batch and a quantitative structural MRI quality index (the Euler number²⁰) using general additive models, before finally being rank-order normalized. For univariate GWASs, these pre-residualized and rank-ordered data were analyzed using a linear model in PLINK v1.9, with a MAF filter of 0.005.

To aid in the evaluation and interpretation of these univariate results, we have now added a new set of Supplementary Figures (4-6), showing Manhattan and QQ-plots (with LDSC intercepts) for all univariate results.

We have also added some sentences to the main text (Results, page 5, lines 189-197), referring to these new Supplementary figures (**additions marked in red**):

«To validate our analysis approach, we computed genetic correlations (using LD-score regression, LDSC²⁷) between univariate GWAS results on the comparison features (see Manhattan- and QQ-plots in Supplementary Figure 4) and previously published neuroimaging GWAS studies on these brain phenotypes. Results showed a mean r_g of .90 (range: .80-.99, see Supplementary Data 3). For the 23 cerebellar morphological features, we used the univariate GWAS summary statistics (see Supplementary Figures 5-6 for Manhattan- and QQ-plots) to compute genetic correlations between discovery ($n = 27,302$) and replication ($n = 11,264$) samples using LDSC²⁷. These genetic correlations were high (mean r_G : .92; range: .83-1), indicating reliable genetic signals (see Supplementary Data 4).»

4. The univariate GWAS on total cerebellar grey matter and cerebral volumes were performed in PLINK. Which model was used, which covariates? Again, what was the genomic inflation factor and LDSC intercept?

Response: Univariate GWASs of total cerebellar grey matter and cerebral brain phenotypes (pre-residualized and rank-ordered) were conducted in PLINK (version 1.9), using a linear model. For total cerebellar volume, the LDSC intercept was 1.0147, i.e., indicative of minimal to no genomic inflation. Similar values were observed for the other comparison brain phenotypes (LDSC intercept range: 0.9963 to 1.0201).

As mentioned in our answer to comment 3 we have now added a new set of Supplementary Figures (4-6) to the manuscript, showing Manhattan and QQ-plots and LDSC intercept values for all univariate analyses. LDSC intercept values have also been added to Supplementary Table 3 (now showing LDSC-based heritability estimates in addition to the heritability estimates computed using GCTA).

5. In the GCTA h2 estimates, how many PCs were included?

Response: All genetic analyses, including GCTA, were run on morphological features that were pre-revisualized for effects of scanner site, sex, age, estimated total intracranial volume, 40 genetic population components, genetic analysis batch and a quantitative structural MRI quality index (the Euler number²⁰) using general additive models, before finally being rank-order normalized. We have now made a minor change to the manuscript (page 5, line 167) to clarify this. This sentence now reads (*addition highlighted in red*):

«Prior to *all* genetic analyses, morphological features were adjusted for effects of scanner site, sex, age, estimated total intracranial volume, 40 genetic population components, genetic analysis batch and a quantitative structural MRI quality index (the Euler number²⁰) using general additive models, before finally being rank-order normalized (see Online Methods for details).»

6. The authors use permutations to show enrichment for “recent” genetic variation (fig 4). The randomly selected SNPs are match for allele frequency with brain related SNPs, but sampling 41,549 random SNPs confined within the MAF bins might be too random. This disregards the correlation structure (LD) between the 41,549 SNPs. SNPs in high LD are likely to have the same “age”. Will the results still hold when the authors select independent SNPs from the GWAS: only top hits or at least stringently LD-pruned SNPs only?

Response: We thank the reviewer for raising this important point. Our motivation for running these analyses on all candidate SNPs was twofold. First, selecting all candidate SNPs provided us with significantly more SNPs, enabling better estimates of SNP age distributions. Second, because of the mentioned LD structure, lead SNPs are not necessarily the most relevant “causal” SNPs, and including all candidate SNPs might thus cover plausible causal variants more broadly. However, disregarding the strong LD among all candidate SNPs is indeed highly likely to bias results, when comparing sets of highly correlated SNPs to null-distributions composed of randomly selected SNPs. We thank the reviewer for pointing this out and we have now rerun the analyses on lead SNPs only (which are mutually independent at an r^2 -threshold of <0.1). While the results change in some respects (e.g., the peak around 300k years ago is no longer significant, although we note a nominally significant peak between 240-260k years ago), they still show significant enrichment for recent evolutionary changes (between 20k and 40k years ago, see new Figure 4A, below).

Figure 4: Lead SNPs associated with cerebellar morphology are enriched for evolutionary recent mutations in the human genome. A: Histogram of estimated SNP age (ranging from 0 to 2 million years, in bins on 20,000 years) for 548 independent lead SNPs associated with cerebellar morphology. The solid black and dotted lines denote the mean and upper 95th confidence interval (Bonferroni corrected across 100 time bins) derived from a null model constructed from 10,000 of equally sized sets of SNPs randomly drawn from the Human Genome Dating Atlas of Variant Age (after matching these to cerebellar lead SNPs in terms of minor allele frequencies). Red bars denote time-bins of significant positive enrichment. **B:** Histogram showing comparative evolutionary enrichment effects for lead SNPs identified in a multivariate GWASs of cerebrocortical area (for cerebrocortical thickness, see Supplementary Figure 11). See Supplementary Data 11-12 for full numerical results.

The much smaller number of lead SNPs (relative to candidate SNPs) does come with another challenge, as numbers of SNPs (both in observed data and null models) approach zero in many time bins. This is especially an issue for the few lead SNPs associated with hippocampal morphology (182 SNPs), and we have consequently removed this comparison phenotype from our analyses. The very low number of SNPs in some time bins also affects the estimated variance for many time bins, artificially inflating z-scores for time-bins with low counts. Hence, we have also removed the z-score comparison figure (former Figure 4B) and replaced this with a histogram for lead SNP derived from a previous multivariate analysis of regional cerebrocortical surface area (the corresponding histogram for regional cerebrocortical thickness can be found as an additional Supplementary Figure 11), also shown below:

Supplementary Figure 11: Figure 4: Lead SNPs associated with cerebrocortical thickness are enriched for evolutionary recent mutations in the human genome: Histogram of estimated SNP age (ranging from 0 to 2 million years, in bins on 20,000 years) for 923 independent lead SNPs associated with regional cerebrocortical thickness. The solid black and dotted lines denote the mean and upper 95th confidence interval (Bonferroni corrected across 100 time-bins) derived from a null model constructed from 10,000 of equally sized sets of SNPs randomly drawn from the Human Genome Dating Atlas of Variant Age (after

matching these to cerebrocortical thickness-related SNPs in terms of minor allele frequencies). Red bars denote time-bins of significant positive enrichment. See Supplementary Data 12 for full numerical results.

Finally, given the relatively low number of independent lead SNPs (range: 548-862 SNPs across phenotypes), we also ran validation analyses using all independent significant SNPs (LD-threshold at $r^2 < 0.6$, range: 1574-2883 SNPs), which yielded very similar results (see below; now added as Supplementary Figure 12, with full numerical results presented in a new Supplementary Table 13).

Supplementary Figure 12: Independent SNPs associated with cerebellar and cerebrocortical morphology are enriched for evolutionary recent mutations in the human genome. A: Histogram of estimated SNP age (ranging from 0 to 2 million years, in bins on 20,000 years) for independent lead SNPs associated with A) cerebellar morphology (1936 SNPs), B) regional cerebrocortical surface area (3606 SNPs) and C) regional cerebrocortical thickness (2993 SNPs). The solid black and dotted lines denote the mean and upper 95th confidence interval (Bonferroni corrected across 100 time-bins) derived from a null model constructed from 10,000 of equally sized sets of SNPs randomly drawn from the Human Genome Dating Atlas of Variant Age (after matching these to brain-phenotype-related SNPs in terms of minor allele frequencies). Red bars denote time-bins of significant positive enrichment. See Supplementary Data 13 for full numerical results.

Given these new results, we have made several changes (*highlighted in red*) to the Results, Discussion and Materials and Methods sections, as detailed below:

Results (pages 11-12, lines 329-415):

«Genetic variants associated with cerebellar morphology are enriched for evolutionary recent mutations in the human genome.

We next mapped the evolutionary age of cerebellar **lead SNPs (LD-threshold at $r^2 < 0.1$)** by merging them with a recently published dataset on dated mutations in the human genome²⁸. Following the analysis procedure established by Libedinsky et al.²⁹, we plotted the histogram of dated **lead SNPs** over the last 2 million years in bins of 20,000 years (Figure 4A and Supplementary Data 11), and tested for positive or negative enrichment by comparing them to empirical null distributions derived from 10,000 randomly drawn and equally sized sets of all SNPs in the full human genome dating dataset (after matching them to **cerebellum-associated lead SNPs** in terms of minor allele frequency; see Online Methods for details).

Figure 4: Lead SNPs associated with cerebellar morphology are enriched for evolutionary recent mutations in the human genome. A: Histogram of estimated SNP age (ranging from 0 to 2 million years, in bins on 20,000 years) for 548 independent lead SNPs associated with cerebellar morphology. The solid black and dotted lines denote the mean and upper 95th confidence interval (Bonferroni corrected across 100 time-bins) derived from a null model constructed from 10,000 of equally sized sets of SNPs randomly drawn from the Human Genome Dating Atlas of Variant Age (after matching these to cerebellum-related SNPs in terms of minor allele frequencies). Red bars denote time-bins of significant positive enrichment. **B:** Histogram showing comparative evolutionary enrichment effects for lead SNPs identified in a multivariate GWAS of cerebrocortical area (for cerebrocortical thickness, see Supplementary Figure 11), See Supplementary Data 11-12 for full numerical results.

Results revealed positive enrichment for cerebellar lead SNPs in the time bin ranging from 20-40,000 years ago, i.e., overlapping the Upper Paleolithic, a period characterized by rapid cultural evolution and the first evidence of several uniquely human behaviors (often referred to as behavioral modernity), such as the recording of information onto objects³⁰.

For comparison, Figure 4B also shows results from an analysis of lead SNPs identified in a previous multivariate GWAS study of regional cerebrocortical surface area³¹, while Supplementary Figure 11 shows the corresponding histogram for regional cerebrocortical thickness. As can be seen, both these cerebrocortical phenotypes also showed significant enrichment in partially overlapping time bins (i.e., 20-60,000 years ago). For full numerical results in Supplementary Data 12. Given the relatively low number of independent lead SNPs (range: 548-862 SNPs across phenotypes), we also ran validation analyses using all independent significant SNPs (LD-threshold $r^2 < 0.6$, range: 1574-2883 SNPs), which yielded very similar results (see Supplementary Figure 12 and Supplementary Table 13).

Discussion (page 20, lines 663-685):

«SNP- and gene-level results from the current study also bolster and refine the notion of relatively recent changes in cerebellar morphology over human evolution³²⁻³⁵. Specifically, we found that lead SNPs associated with cerebellar morphology were enriched for SNPs with an estimated age of 20-40 thousand years. This falls within the Upper Paleolithic (10-50 thousand years ago), a period characterized by rapid cultural evolution, and coinciding with the first evidence of several uniquely human behaviors (often referred to as behavioral modernity), such as the recording of information onto objects³⁰. Converging evidence for changes in brain anatomy around this evolutionary

period comes from the fossil skull record, where only fossils dated less than 35,000 years old fall within the range of shape-variation seen in modern humans^{36,37}. Of note, one key feature of modern skulls is their “globular” shape, in part characterized by an enlarged posterior fossa (the cranial compartment housing the cerebellum)^{36,37}. A similar pattern of enrichment for recent evolutionary time bins (20-60 thousand years ago) was also found when analyzing lead SNPs derived from previous studies on multivariate cerebro-cortical³¹ morphology.

Gene level analyses further showed that genes associated with inter-individual variation in cerebellar morphology are enriched for genes linked to human accelerated regions (HARs)³⁸ of the genome. HARs denote previously conserved regions of the genome that were subject to a burst of changes in humans after the divergence of humans from chimpanzees about 6-8 million years ago³⁸. Of note, a recent GWAS on total cerebellar volume found no enrichment for HAR-linked genes¹⁸, suggesting that SNPs associated with regional cerebellar variation may be driving this effect. **When comparing this finding with results from other multivariate brain phenotypes, we observed** that HAR gene enrichment was nominally stronger for cerebellar morphology than for vertex-wise cerebrocortical thickness and area, and significantly stronger than for hippocampal regional volumes (Figure 5D).»

Materials and Methods (pages 31-32, lines 1116-1138):

«Analysis of Human Dating Genome Data. The Atlas of Variant Age for chromosomes 1-22 was downloaded from the Human Genome Dating (HGD) website: <https://human.genome.dating/>. This atlas contains more than 45 million SNPs which has been assigned dates of origin based on a recombination clock and mutation clock applied to two large-scale sequencing datasets (1000 Genomes Project³⁹ and The Simons Genome Diversity Project⁴⁰), with no assumptions made about demographic or selective processes²⁸. The current study used the median joint age estimates from both clocks when analyzing SNPs present in both datasets in combination (i.e., 13,694,493 SNPs).

After merging dated SNPs with the 40,405,505 SNPs also present in the Haplotype Reference Consortium reference data (to add minor allele frequencies, MAF) and removing 715,083 (5.2%) SNPs with missing MAF values as well as the very few (14,549, 0.1%) SNPs dated older than 2 million years, 12,960,066 SNPs remained.

548 of these dated SNPs were matched to lead SNPs linked to cerebellar morphology (defined as being in mutual LD at threshold $r^2 < 0.1$), hereafter referred to as cerebellar-SNPs. Partially because very rare variants (Minor Allele Frequency (MAF) < 0.005) had been removed prior to the multivariate GWAS analysis, MAF was not equally distributed between these cerebellar-SNPs and the full range of dated SNPs in the HGD dataset. Importantly, MAF has been shown to be systematically related to the estimated age of SNPs, with a higher proportion of low-MAF SNPs in more recent than in older time-bins^{28,29}. Consequently, following the analysis approach established by Libedinsky et al.²⁹, we first determined the MAF-distribution of cerebellar-SNPs (across eight bins: <0.0001 ; $0.0001-0.001$;

0.001-0.01; 0.01-0.1; 0.1-0.2; 0.2-0.3; 0.3-0.4; 0.4-0.5) and selected random set of 2,892,270 SNPs from the HGD dataset that were matched to the cerebellar-SNPs in terms of MAF-bin distribution. Similar MAF-matched HDG subsets were created for lead SNPs associated with the two cerebrocortical comparison phenotypes, i.e., regional cerebrocortical surface area and thickness (862 and 714 lead SNPs, respectively).

For statistical inference we constructed null models (separate for each brain phenotype) by randomly drawing sets of SNPs (of equal size to the number of phenotype-linked lead SNPs) from the MAF-matched HGD-datasets and computing the histograms of estimated dates from 0 to 2 million years ago (divided into 100 bins of 20,000 years) over 10,000 iterations. From these null models we extracted bin means as well as significance thresholds (defined as the upper and lower 99.95th percentile of the null model (i.e. corresponding to a **two-tailed** threshold of 0.05 **Bonferroni corrected** across 100 bins).

For the validation analyses, we ran the same analyses based on the larger number (range across phenotypes: 1574-2883) of individual significant SNPs (defined as being in mutual LD at threshold $r^2 < 0.6$).»

7. It could be interesting to run single cell enrichment analyses using FUMA as well: <https://fuma.ctglab.nl/celltype>.

Response: Yes, this is indeed an interesting idea that we considered pursuing while analyzing the data. Unfortunately, however, to the best of our knowledge there exists only very few available single cell datasets derived from cerebellar tissue. Given the very different gene expression pattern seen in the cerebellum relative to many other brain regions, we thus decided against this idea. However, given the high likelihood of more cerebellar single cell expression data becoming available (e.g., <https://www.biorxiv.org/content/10.1101/2022.10.12.511898v1.full.pdf>) this is an excellent idea for follow-up studies.

8. The authors use conditional/conjunctional FDR to assess and leverage a genetic overlap between psychiatric diseases and the cerebellum. Can't this be biased due to SNP LD-score? Those SNPs with high LD scores are more likely to have higher Chisq statistics for both traits if they tag more genetic variation and thus conditional QQ-plots show the inflation pattern.

Response: The conditional/conjunctional FDR analyses deal with potential biases due to LD-structure using three strategies⁴¹. First, large LD-blocks with complex LD -structure (e.g., the MHC-complex on Chromosome 6) are removed prior to analysis. Second, we controlled for spurious enrichment by calculating all conditional Q-Q plots after random pruning averaged over 500 iterations. At each iteration, one SNP in every LD block (defined by an $r^2 > 0.1$) was randomly selected and the empirical cumulative distributions were computed using the corresponding p-values. Third, we only report lead SNPs after clumping (using an r^2 threshold of < 0.1 ^{41,42}).

The second of these strategies was not mentioned in the previous version of the manuscript, and we have now added the following sentence to the

Materials and Methods section (page 33, lines 1200-1203) to better clarify this aspect of the conditional/conjunctive analyses:

“We further controlled for spurious enrichment by calculating all conditional Q-Q plots after random pruning averaged over 500 iterations. At each iteration, one SNP in every LD block (defined by $r^2 > 0.1$) was randomly selected and the empirical cumulative distributions were computed using the corresponding p-values.”

9. Will the authors release the univariate and multivariate summary statistics for the public?

Response: Yes, both multivariate and univariate summary statistics will be made available both in the GWAS Catalog [<https://www.ebi.ac.uk/gwas/>] and in our github repository [<https://github.com/norment/open-science>] upon publication of the manuscript.

Minor points:

11. Fig 3: there is no colour legend for the colours in the cerebellum (Z-scores probably?).

Response: We thank the reviewer for pointing out this unfortunate omission. A color legend has now been added to Fig. 3 (below), as well as to Supplementary Figures 7 and 8 (see revised manuscript).

Figure 3: Loci lead SNPs show spatially heterogeneous and replicable effects across the cerebellar cortex. The 351 loci lead SNPs identified by MOSTest are plotted as a function of

main overall effect across all cerebellar features (x-axis: mean Z-score) and most extreme effect for any single cerebellar feature (y-axis: most extreme Z-score across features), and colour coded by SNP discovery method. The cerebellar flat-maps show discovery (left) and replication (right) sample regional distributions of Z-scores (color-scale range from -10 to 10) for a few selected lead SNPs (see Supplementary Data 10 for individual feature Z-scores for all 351 discovery sample loci lead SNPs). SNP rs7877685 was only present in the discovery sample.

12. Fig 5C: developmental stages could be ordered from first to last instead of -log₁₀(P) to make it more intuitive.

Response: We fully agree that the suggested chronological ordering would improve the figure and have revised it accordingly in the revised manuscript, see also the revised Figure 5 below:

Figure 5: Gene mapping reveals selective enrichment for cerebellar and prenatal brain tissue, as well as for genes linked to human accelerated regions. **A:** MAGMA mapped the full range of SNPs from the multivariate GWAS to 19,329 protein-coding genes, 534 of which were genome-wide significant (Bonferroni corrected threshold marked with red line), with the 30 most significant genes labeled in the gene-based Manhattan plot. **B-C:** MAGMA gene-property analyses revealed selective brain expression in the cerebellum and during prenatal developmental stages. **D:** MAGMA gene-set analysis revealed significant enrichment for sets of genes previously linked to human accelerated regions (HARs). Figure 4d also shows results from comparative analyses using summary statistics from other multivariate GWASs of MRI-based brain, as well as significant results from statistical tests comparing beta-weights for HAR-linked genes across multivariate brain features. Horizontal red lines mark the Bonferroni-corrected significance threshold for each subplot. See Supplementary Data 14-17 for full results.

References:

1. van der Meer, D., Frei, O., Kaufmann, T., Shadrin, A.A., Devor, A., Smeland, O.B., Thompson, W.K., Fan, C.C., Holland, D., Westlye, L.T., et al. (2020). Understanding the genetic determinants of the brain with MOSTest. *Nat Commun* *11*, 3512. 10.1038/s41467-020-17368-1.
2. King, M., Hernandez-Castillo, C.R., Poldrack, R.A., Ivry, R.B., and Diedrichsen, J. (2019). Functional boundaries in the human cerebellum revealed by a multi-domain task battery. *Nat Neurosci* *22*, 1371-1378. 10.1038/s41593-019-0436-x.
3. Nassar, R., Kaczkurkin, A.N., Xia, C.H., Sotiras, A., Pehlivanova, M., Moore, T.M., Garcia de La Garza, A., Roalf, D.R., Rosen, A.F.G., Lorch, S.A., et al. (2019). Gestational Age is Dimensionally Associated with Structural Brain Network Abnormalities Across Development. *Cereb Cortex* *29*, 2102-2114. 10.1093/cercor/bhy091.
4. Pehlivanova, M., Wolf, D.H., Sotiras, A., Kaczkurkin, A.N., Moore, T.M., Ciric, R., Cook, P.A., Garcia de La Garza, A., Rosen, A.F.G., Ruparel, K., et al. (2018). Diminished Cortical Thickness Is Associated with Impulsive Choice in Adolescence. *J Neurosci* *38*, 2471-2481. 10.1523/JNEUROSCI.2200-17.2018.
5. Sotiras, A., Resnick, S.M., and Davatzikos, C. (2015). Finding imaging patterns of structural covariance via Non-Negative Matrix Factorization. *Neuroimage* *108*, 1-16. 10.1016/j.neuroimage.2014.11.045.
6. Sotiras, A., Toledo, J.B., Gur, R.E., Gur, R.C., Satterthwaite, T.D., and Davatzikos, C. (2017). Patterns of coordinated cortical remodeling during adolescence and their associations with functional specialization and evolutionary expansion. *Proc Natl Acad Sci U S A* *114*, 3527-3532. 10.1073/pnas.1620928114.
7. Varikuti, D.P., Genon, S., Sotiras, A., Schwender, H., Hoffstaedter, F., Patil, K.R., Jockwitz, C., Caspers, S., Moebus, S., Amunts, K., et al. (2018). Evaluation of non-negative matrix factorization of grey matter in age prediction. *Neuroimage* *173*, 394-410. 10.1016/j.neuroimage.2018.03.007.
8. E, K.-H., Chen, S.-H.A., Ho, M.-H.R., and Desmond, J.E. (2014). A meta-analysis of cerebellar contributions to higher cognition from PET and fMRI studies. *Hum Brain Mapp*.
9. Stoodley, C.J., and Schmahmann, J.D. (2009). Functional topography in the human cerebellum: a meta-analysis of neuroimaging studies. *NeuroImage*.
10. Buckner, R.L., Krienen, F.M., Castellanos, A., Diaz, J.C., and Yeo, B.T.T. (2011). The organization of the human cerebellum estimated by intrinsic functional connectivity. *Journal of Neurophysiology*.
11. Sang, L., Qin, W., Liu, Y., Han, W., Zhang, Y., Jiang, T., and Yu, C. (2012). Resting-state functional connectivity of the vermal and hemispheric subregions of the cerebellum with both the cerebral cortical networks and subcortical structures. *NeuroImage*. Elsevier Inc.
12. Krienen, F.M., and Buckner, R.L. (2009). Segregated fronto-cerebellar circuits revealed by intrinsic functional connectivity. *Cerebral Cortex*.
13. Bernard, J.A., Seidler, R.D., Hassevoort, K.M., Benson, B.L., Welsh, R.C., Wiggins, J.L., Jaeggi, S.M., Buschkuhl, M., Monk, C.S., Jonides, J., and Peltier, S.J. (2012). Resting state cortico-cerebellar functional connectivity networks: a comparison of anatomical and self-organizing map approaches. *Front Neuroanat*.

14. Kipping, J.A., Grodd, W., Kumar, V., Taubert, M., Villringer, A., and Margulies, D.S. (2013). Overlapping and parallel cerebello-cerebral networks contributing to sensorimotor control: an intrinsic functional connectivity study. *NeuroImage*.
15. Wang, C., Kipping, J., Bao, C., Ji, H., and Qiu, A. (2016). Cerebellar Functional Parcellation Using Sparse Dictionary Learning Clustering. *Front Neurosci* 10, 188. 10.3389/fnins.2016.00188.
16. Kipping, J.A., Tuan, T.A., Fortier, M.V., and Qiu, A. (2016). Asynchronous Development of Cerebellar, Cerebello-Cortical, and Cortico-Cortical Functional Networks in Infancy, Childhood, and Adulthood. *Cereb Cortex*. 10.1093/cercor/bhw298.
17. Loughnan, R.J., Shadrin, A.A., Frei, O., van der Meer, D., Zhao, W., Palmer, C.E., Thompson, W.K., Makowski, C., Jernigan, T.L., Andreassen, O.A., et al. (2022). Generalization of cortical MOSTest genome-wide associations within and across samples. *Neuroimage* 263, 119632. 10.1016/j.neuroimage.2022.119632.
18. Tissink, E., de Lange, S.C., Savage, J.E., Wightman, D.P., de Leeuw, C.A., Kelly, K.M., Nagel, M., van den Heuvel, M.P., and Posthuma, D. (2022). Genome-wide association study of cerebellar volume provides insights into heritable mechanisms underlying brain development and mental health. *Commun Biol* 5, 710. 10.1038/s42003-022-03672-7.
19. Alnaes, D., Kaufmann, T., Marquand, A.F., Smith, S.M., and Westlye, L.T. (2020). Patterns of sociocognitive stratification and perinatal risk in the child brain. *Proc Natl Acad Sci U S A* 117, 12419-12427. 10.1073/pnas.2001517117.
20. Rosen, A.F.G., Roalf, D.R., Ruparel, K., Blake, J., Seelaus, K., Villa, L.P., Ciric, R., Cook, P.A., Davatzikos, C., Elliott, M.A., et al. (2018). Quantitative assessment of structural image quality. *Neuroimage* 169, 407-418. 10.1016/j.neuroimage.2017.12.059.
21. Alfaro-Almagro, F., McCarthy, P., Afyouni, S., Andersson, J.L.R., Bastiani, M., Miller, K.L., Nichols, T.E., and Smith, S.M. (2021). Confound modelling in UK Biobank brain imaging. *Neuroimage* 224, 117002. 10.1016/j.neuroimage.2020.117002.
22. Goeman, J.J., and Buhlmann, P. (2007). Analyzing gene expression data in terms of gene sets: methodological issues. *Bioinformatics* 23, 980-987. 10.1093/bioinformatics/btm051.
23. de Leeuw, C.A., Mooij, J.M., Heskes, T., and Posthuma, D. (2015). MAGMA: generalized gene-set analysis of GWAS data. *PLoS Comput Biol* 11, e1004219. 10.1371/journal.pcbi.1004219.
24. Watanabe, K., Taskesen, E., van Bochoven, A., and Posthuma, D. (2017). Functional mapping and annotation of genetic associations with FUMA. *Nat Commun* 8, 1826. 10.1038/s41467-017-01261-5.
25. Okbay, A., Wu, Y., Wang, N., Jayashankar, H., Bennett, M., Nehzati, S.M., Sidorenko, J., Kweon, H., Goldman, G., Gjorgjieva, T., et al. (2022). Polygenic prediction of educational attainment within and between families from genome-wide association analyses in 3 million individuals. *Nat Genet* 54, 437-449. 10.1038/s41588-022-01016-z.
26. Wang, Q., Dhindsa, R.S., Carss, K., Harper, A.R., Nag, A., Tachmazidou, I., Vitsios, D., Deevi, S.V.V., Mackay, A., Muthas, D., et al. (2021). Rare variant contribution to human disease in 281,104 UK Biobank exomes. *Nature* 597, 527-532. 10.1038/s41586-021-03855-y.
27. Bulik-Sullivan, B.K., Loh, P.R., Finucane, H.K., Ripke, S., Yang, J., Schizophrenia Working Group of the Psychiatric Genomics, C., Patterson, N., Daly, M.J., Price, A.L., and Neale,

- B.M. (2015). LD Score regression distinguishes confounding from polygenicity in genome-wide association studies. *Nat Genet* 47, 291-295. 10.1038/ng.3211.
28. Albers, P.K., and McVean, G. (2020). Dating genomic variants and shared ancestry in population-scale sequencing data. *PLoS Biol* 18, e3000586. 10.1371/journal.pbio.3000586.
 29. Libedinsky, I., Wei, Y., Leeuw, C.d., Rilling, J., Posthuma, D., and Heuvel, M.P.v.d. (2023). Genetic timeline of human brain and cognitive traits. *bioRxiv*, 2023.2002.2005.525539. 10.1101/2023.02.05.525539.
 30. Davies, S. (2021). Behavioral Modernity in Retrospect. *Topoi* 40, 221-232. 10.1007/s11245-019-09671-4.
 31. van der Meer, D., Kaufmann, T., Shadrin, A.A., Makowski, C., Frei, O., Roelfs, D., Monereo-Sanchez, J., Linden, D.E.J., Rokicki, J., Alnaes, D., et al. (2021). The genetic architecture of human cortical folding. *Sci Adv* 7, eabj9446. 10.1126/sciadv.abj9446.
 32. Barton, R.A., and Venditti, C. (2014). Rapid Evolution of the Cerebellum in Humans and Other Great Apes. *Curr. Biol*.
 33. Macri, S., Savriama, Y., Khan, I., and Di-Poi, N. (2019). Comparative analysis of squamate brains unveils multi-level variation in cerebellar architecture associated with locomotor specialization. *Nat Commun* 10, 5560. 10.1038/s41467-019-13405-w.
 34. Smaers, J.B., Turner, A.H., Gomez-Robles, A., and Sherwood, C.C. (2018). A cerebellar substrate for cognition evolved multiple times independently in mammals. *Elife* 7. 10.7554/eLife.35696.
 35. Harrison, P.W., and Montgomery, S.H. (2017). Genetics of Cerebellar and Neocortical Expansion in Anthropoid Primates: A Comparative Approach. *Brain Behav Evol* 89, 274-285. 10.1159/000477432.
 36. de Sousa, A.A., Beaudet, A., Calvey, T., Bardo, A., Benoit, J., Charvet, C.J., Dehay, C., Gomez-Robles, A., Gunz, P., Heuer, K., et al. (2023). From fossils to mind. *Commun Biol* 6, 636. 10.1038/s42003-023-04803-4.
 37. Neubauer, S., Hublin, J.J., and Gunz, P. (2018). The evolution of modern human brain shape. *Sci Adv* 4, eaao5961. 10.1126/sciadv.aao5961.
 38. Wei, Y., de Lange, S.C., Scholtens, L.H., Watanabe, K., Ardesch, D.J., Jansen, P.R., Savage, J.E., Li, L., Preuss, T.M., Rilling, J.K., et al. (2019). Genetic mapping and evolutionary analysis of human-expanded cognitive networks. *Nat Commun* 10, 4839. 10.1038/s41467-019-12764-8.
 39. Genomes Project, C., Auton, A., Brooks, L.D., Durbin, R.M., Garrison, E.P., Kang, H.M., Korbel, J.O., Marchini, J.L., McCarthy, S., McVean, G.A., and Abecasis, G.R. (2015). A global reference for human genetic variation. *Nature* 526, 68-74. 10.1038/nature15393.
 40. Mallick, S., Li, H., Lipson, M., Mathieson, I., Gymrek, M., Racimo, F., Zhao, M., Chennagiri, N., Nordenfelt, S., Tandon, A., et al. (2016). The Simons Genome Diversity Project: 300 genomes from 142 diverse populations. *Nature* 538, 201-206. 10.1038/nature18964.
 41. Smeland, O.B., Frei, O., Shadrin, A., O'Connell, K., Fan, C.C., Bahrami, S., Holland, D., Djurovic, S., Thompson, W.K., Dale, A.M., and Andreassen, O.A. (2020). Discovery of shared genomic loci using the conditional false discovery rate approach. *Hum Genet* 139, 85-94. 10.1007/s00439-019-02060-2.
 42. Andreassen, O.A., Thompson, W.K., Schork, A.J., Ripke, S., Mattingsdal, M., Kelsoe, J.R., Kendler, K.S., O'Donovan, M.C., Rujescu, D., Werge, T., et al. (2013). Improved

detection of common variants associated with schizophrenia and bipolar disorder using pleiotropy-informed conditional false discovery rate. *PLoS Genet* 9, e1003455. [10.1371/journal.pgen.1003455](https://doi.org/10.1371/journal.pgen.1003455).

Reviewers' comments:

Reviewer #1 (Remarks to the Author):

The revised version of the manuscript "The genetic architecture of human cerebellar morphology supports a key role for the cerebellum in human evolution and psychopathology" by Moberget and colleagues reads well and answered most of the questions I had while reading the first version. I only have a few minor points left.

Response: *We thank the reviewer for this positive assessment of our revised manuscript and for the additional suggestions for further improvements.*

1) In answer to my previous point 1 the authors do explain why they chose to investigate the cerebellum through multivariate analysis (essentially, to pick up moderate bidirectional effects), the introductory paragraph ("Of note.." still discusses the separate parts of the cerebellum with respect to their separate functions). Please add a sentence that justifies the multivariate approach to help the reader.

Response: *We agree that our motivations for choosing a multivariate analysis approach was not made sufficiently clear in the Introduction, and have now revised this paragraph, which now reads (additions marked in red):*

"Of note, the few existing cerebellar genome-wide association studies (GWAS) have mostly been restricted to total cerebellar volume^{1,2}, thus largely ignoring regional variation in cerebellar morphology. Importantly, such variation in the relative volumes of cerebellar subregions (i.e., variation in cerebellar shape independent of total cerebellar volume) has been associated with variation in behavioral repertoires in several species^{3,4}, including domain-general cognition in primates⁴. **Further, several recent studies investigating other brain phenotypes (e.g. the cerebral cortex^{5,6} and the hippocampus⁷) have shown that genetic signals tends to be distributed across brain regions, rendering multivariate methods more sensitive than univariate methods.**

We **thus** here performed a multivariate GWAS of MRI-derived regional cerebellar morphological features in a large population-based sample from the UK biobank (n discovery = 27,302; n replication = 11,264), functionally characterized the genetic signal, tested for enrichment of SNPs and genes linked to human evolution, and assessed genetic overlap with major mental disorders."

2) I do appreciate the addition of the univariate results for the cerebellar covariance patterns, but I do still wonder about the addition of the "comparison phenotypes". The authors explain their rationale better than the in the previous version of this manuscript, but I feel it could be even further simplified.

"To validate our analysis approach, we computed genetic correlations (using LD-score regression, LDSC32) between univariate GWAS results on the comparison features (see Manhattan and QQ-plots in Supplementary Figure 5) and previously published neuroimaging GWAS studies on these brain phenotypes.

If I am totally honest, I still do not see the purpose of this analysis. The fact that these overlap could stem from the fact that many of the previously published data also includes (part of) the UKB data and this does not validate the approach for the

main paper. Why not leave out all the phenotypes that are not truly used (as far as I can see, the phenotypes that are explicitly compared to the multivariate analysis are hippocampus, Surface Area and Cortical thickness)?

Response: *We agree that our full motivation for including all the comparison brain phenotypes was not sufficiently clear in the previous version of the manuscript. To clarify, our motivation was two-fold: First, by computing genetic correlations between results from the current and previous studies, we were able to assess the overall consistency of our findings with the existing literature. While we agree with the reviewer that these results are not in themselves novel, we nonetheless believe they provide important sanity checks for our analyses and results and have thus decided to keep these results in the revised manuscript.*

Second, the comparison phenotypes (univariate as well as multivariate) help to assess the relative strength and/or specificity of any cerebellar results. While we believe this was clear with respect to the multivariate phenotypes (i.e., regional cortical thickness and area, as well as regional hippocampal volumes), we agree with the reviewer that this motivation was not made explicit enough with respect to the univariate comparison phenotypes. We have consequently made the following changes to the text introducing the comparison brain phenotypes (page 5, lines 166-169, additions marked in red):

«Our motivation for including these comparison phenotypes was twofold; first to allow for methodological sanity checks by comparing current results to previous publications, and second, to assess the relative strength and/or specificity of any cerebellar results.»

3) Around line 205: Maybe add a sentence on that/how the univariate GWAS were run; the current description jumps from preprocessing the phenotypes to SNP heritability.

Response: *We agree that this section could indeed be improved by providing a bit more detail. In the revised version, this section now reads as follows (additions marked in red):*

«Prior to all genetic analyses, morphological features were adjusted for effects of scanner site, sex, age, estimated total intracranial volume, 40 genetic population components, genetic analysis batch and a quantitative structural MRI quality index (the Euler number⁸) using general additive models, before finally being rank-order normalized (see Online Methods for details).

To validate our analysis approach, we computed genetic correlations (using LD-score regression, LDSC⁹) between univariate GWAS results (computed using a linear model in PLINK 1.9) on the comparison features (see Manhattan- and QQ-plots in Supplementary Figure 5) and previously published neuroimaging GWAS studies on these brain phenotypes.»

4) Line 236: Supplementary Data 8 should probably read Supplementary Data 6.

Response: *We checked – and Supplementary Data 8 is indeed correct.*

5) Supplementary Data: The table descriptions contain some typos e.g. thickness (SI6) mapping strategies (SI22-25).

Response: Thanks for noticing these typos, which have now been fixed. We have also thoroughly inspected the rest of the manuscript and supplementary information for any remaining typos. We identified the following, which have now been fixed:

- In SI-Data 36-37 full references to the used disorder GWAS studies were missing – this has now been fixed.

Reviewer #2 (Remarks to the Author):

I thank the authors for their comprehensive reply and thorough additional analyses. I have no further remaining questions or remarks.

Response: We thank the reviewer for this positive evaluation of our current revision, and for the many constructive suggestions in the previous round.

References:

1. Chambers, T. *et al.* Genetic common variants associated with cerebellar volume and their overlap with mental disorders: a study on 33,265 individuals from the UK-Biobank. *Mol Psychiatry* **27**, 2282-2290 (2022).
2. Tissink, E. *et al.* Genome-wide association study of cerebellar volume provides insights into heritable mechanisms underlying brain development and mental health. *Commun Biol* **5**, 710 (2022).
3. Macri, S., Savriama, Y., Khan, I. & Di-Poi, N. Comparative analysis of squamate brains unveils multi-level variation in cerebellar architecture associated with locomotor specialization. *Nat Commun* **10**, 5560 (2019).
4. Smaers, J.B., Turner, A.H., Gomez-Robles, A. & Sherwood, C.C. A cerebellar substrate for cognition evolved multiple times independently in mammals. *Elife* **7**(2018).
5. van der Meer, D. *et al.* Understanding the genetic determinants of the brain with MOSTest. *Nat Commun* **11**, 3512 (2020).
6. van der Meer, D. *et al.* The genetic architecture of human cortical folding. *Sci Adv* **7**, eabj9446 (2021).
7. Bahrami, S. *et al.* Distributed genetic architecture across the hippocampal formation implies common neuropathology across brain disorders. *Nat Commun* **13**, 3436 (2022).
8. Rosen, A.F.G. *et al.* Quantitative assessment of structural image quality. *Neuroimage* **169**, 407-418 (2018).
9. Bulik-Sullivan, B.K. *et al.* LD Score regression distinguishes confounding from polygenicity in genome-wide association studies. *Nat Genet* **47**, 291-5 (2015).

Dear editorial team,

Thank you for notifying us about the needed changes. Please find our responses to your specific points below. And see the marked up manuscript file (with track changes) for all changes from the previous version.

Specific points:

-Thank you for depositing your summary statistics on the GWAS Catalog, though we cannot see them from searching the database (GCST90728589). Please provide a confirmation email or similar documentation from the GWAS Catalog confirming that this accession is indeed correct.

- We have now contacted GWAS catalog and asked them to lift the moratorium on the GWAS summary statistics files. These should now be available within 24 hours.

-Given the amount of time between your last submission, please double-check that the description of novel variants (228) here is still accurate, and outline any changes to the text in a brief cover letter and marked-up manuscript file.

- We agree that this is a very important point and have identified and read recent publications that could provide more updated data on identified loci related to cerebellar morphology. We identified 3 publications that examine the genetic architecture of cerebellar morphology and that have been published within the last year or two. These are discussed below.
1. Carrión-Castillo, A., Boeckx, C. Insights into the genetic architecture of cerebellar lobules derived from the UK Biobank. *Sci Rep* **14**, 9488 (2024). <https://doi.org/10.1038/s41598-024-59699-9>
 - This publication uses the same GWAS summary statistics data as reported in Smith et al., which are already included in the current version of the manuscript. It thus does not provide new data on variants and has not been examined further.
 2. Liang, Y., Ma, D., Li, M. *et al.* Exome sequencing identifies novel genes associated with cerebellar volume and microstructure. *Commun Biol* **8**, 344 (2025). <https://doi.org/10.1038/s42003-025-07797-3>
 - This publication uses exome-sequencing data (in contrast to the more common genotyping arrays used in the GWAS reported in the current manuscript), and could thus potentially have discovered variants or loci

reported as novel in the current version of the manuscript. We thus retrieved reported significant variants from this publication and compared them to the loci reported in the current manuscript. While the Liang et al paper report some variants that overlap 2 loci also reported in our publication, none of these loci were reported by us as novel (since they have also been reported in previous publications). These results thus do not have an impact on the novelty description in the current manuscript.

3. Primus S., *et al.* Beyond volume: Unraveling the genetics of human brain geometry. *Sci. Adv.* **11**, eadr1644(2025). DOI:[10.1126/sciadv.adr1644](https://doi.org/10.1126/sciadv.adr1644)
 - This publication assesses *cerebellar shape*, rather than volume, and is thus not directly comparable to the current manuscript of the previous GWAS studies on cerebellar morphology. However, shape is clearly an important feature of brain morphology beyond volumes, and the data-driven decomposition and multivariate analysis approach taken in our paper should also be sensitive to inter-individual shape differences. The recent paper by Primus et al could thus potentially have discovered novel variants or loci related to this aspect of brain morphology that are reported as novel in the current version of our manuscript. We consequently retrieved reported significant variants from this publication and compared them to the loci reported as novel in the current manuscript. Indeed, two loci that we originally reported as novel in our manuscript overlapped genetic variants that were reported as significantly associated with cerebellar shape in this publication. We have consequently reduced the number of loci claimed as novel from 228 to 226, and inserted a new sentence (and reference to the Primus et al paper) in the results section (page 7, lines 839-841). This now reads (added/alterd text in red):

“Two additional genetic loci identified in the current study were also reported in a recent study investigating genetic associations with cerebellar shape⁴⁶. Thus, 226 of the 351 (i.e., 64%) genetic loci reported here are novel to the literature on of cerebellar morphology genetics (see Supplementary Data 10.

- Please include the city name in affiliations 3.

- This has now been corrected.

- Please supply your Supplementary Information file in PDF format.

- We have now created a new pdf-file that provides an overview of all Supplementary Tables in the Excel-file.

- Please ensure that each Supplementary Data (Supplementary Data 13, 15,16)file supplied is individually cited in the main manuscript text or the Data availability statement (i.e. under a section that will be retained by our typesetters).

- We have now gone through all Supplementary data tables (and references to these in the text). We noticed, and corrected, a few errors.

Your paper has been placed back in the 'Author Approval Folder', which you may access via the following link:

<https://mts-commsbio.nature.com/cgi-bin/main.plex?el=A6Cx7HQS7D7BYEh3F3A9ftdZWIkNPWY6AOXBUCr6l1PkQZ>

Please remedy the point(s) specified above and resubmit your paper following the same steps as before.

If you have any questions, please do not hesitate to contact us.

Best Regards,

Manuscript Administration
Communications Biology